# Neptune: The Long Orbit to Benchmarking Long Video Understanding

## Abstract

This paper describes a semi-automatic pipeline to generate challenging question-answer-decoy sets for understanding long videos. Many existing video datasets and models are focused on short clips (10s-30s). While some long video datasets do exist, they can often be solved by powerful image models applied per frame (and often to very few frames) in a video, and are usually manually annotated at high cost. In order to mitigate both these problems, we propose a scalable dataset creation pipeline which leverages large models (VLMs and LLMs), to automatically generate dense, time-aligned video captions, as well as tough question answer decoy sets for video segments (up to 15 minutes in length). Our dataset Neptune covers a broad range of long video reasoning abilities and consists of a subset that emphasizes multimodal reasoning. Since existing metrics for open-ended question answering are either rule-based or may rely on proprietary models, we provide a new open source model-based metric (GEM) to score open-ended responses on Neptune. Benchmark evaluations reveal that current open-source long video models perform poorly on Neptune, particularly on questions testing temporal ordering, counting and state changes. Through Neptune, we aim to spur the development of more advanced models capable of understanding long videos.

## 1 Introduction

Videos are experiencing an *explosion* moment online, with new research constantly pushing the frontier for video and language tasks such as video question answering (VideoQA) (Xu et al., 2017; Zhong et al., 2022; Xiao et al., 2021; Yang et al., 2021; Mangalam et al., 2023). Early video and language models, while adept at VideoQA, have largely focused on short, trimmed clips (less than 1 minute long (Yu et al., 2019a; Xiao et al., 2021)). The recent release of powerful, longer context multimodal models (eg. Gemini 1.5 (Reid et al., 2024) and GPT4 (Achiam et al., 2023)), however, has ushered in the promise of models being able to reason over millions of tokens, covering longer stretches of videos (many minutes long).

While promising, these claims are often evidenced by qualitative examples, or results on small-size datasets – for example the 1H-VideoQA (Reid et al., 2024) benchmark, which while valuable, only consists of 125 questions. Popular video benchmarks for question answering still tend to focus on short, trimmed clips (*e.g.*, Next-QA (Xiao et al., 2021)). Other datasets that *do* contain longer videos are often 'short-term' benchmarks disguised as long-term ones, evidenced by models that are able to solve them with a single (or a few) frames (*e.g.* some tasks on the LVU dataset (Wu & Krahenbuhl, 2021) such as scene prediction of movies). Other long video datasets may contain strong linguistic biases in multiple choice evaluation, as shown by MoreVQA (Min et al., 2024), which gets strong performance on EgoSchema (Mangalam et al., 2023) without access to the video at all, or can be solved with external internet knowledge, such as those made from popular movies (Li et al., 2023d).

A key challenge in creating a truly long form video understanding dataset is the significant manual cost required to select, watch, understand and annotate long videos with free-form natural language. Answering challenging questions about longer videos is often a *multimodal* (as it may involve listening to the audio track in addition to watching the video), and *non-sequential* endeavour (as sometimes it is necessary to rewind and rewatch key parts to answer a question). Proposing suitable high-level questions that are not trivially solved by a few frames is also tricky for humans to do consistently and with adequate diversity. The key aim of this paper is to solve this challenge by

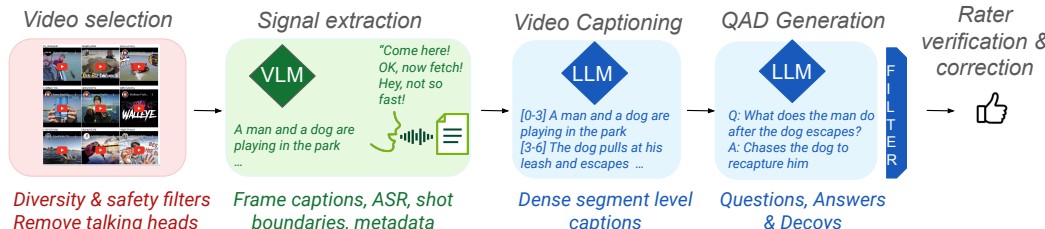

Figure 1: **Pipeline Overview:** Our pipeline consists of 5 key stages - (i) Video selection, where suitable videos are identified from YouTube, (ii) Signal extraction, (iii) Video level captioning, (iv) Question, answer and decoy (QAD) generation and (v) Manual rater verification. The first four stages are entirely automatic. Before rater verification, we automatically filter out QADs that can be solved by an LLM without access to the video content.

leveraging automatic tools to reduce rater effort while at the same retaining quality. Inspired by EgoSchema, we do this by proposing a scalable dataset creation pipeline (Fig. 1) that leverages strong foundational Video Language Models (VLMs) and Large Language Models (LLMs) with carefully designed prompts. We first generate dense, time-aligned video captions automatically, from which tough question-answer-decoy (QAD) sets can be automatically derived. This is done by extracting image captions, automatic speech recognition (ASR), shot boundaries and video metadata, and combining these signals with multi-stage, chain of thought prompting of an LLM. Our pipeline can be applied to any video on YouTube (Fig. 1).

While most of the pipeline is automatic, a comprehensive rater verification stage at the end ensures quality. While other dataset pipelines that are entirely manual (Zhou et al., 2024; Fang et al., 2024; Wang et al., 2024), our verification stage is lightweight, which we show by ablating the automatic part of the pipeline, and measuring the time taken by raters to propose QAs for videos from scratch. Results show that our semi-automatic pipeline almost halves rater effort. Our dataset is called Neptune[1], and covers a diverse range of videos, is multimodal (requires audio and visual information), and poses challenging questions for videos that test a variety of reasoning abilities over long time horizons. Neptune allows for two modes of evaluation: multiple-choice and open-ended question answering. Since existing metrics for open-ended question answering are either rule-based and derived from captioning (WUPS (Wu & Palmer, 1994), CIDEr (Vedantam et al., 2015), etc) or are LLM-based evals that rely on proprietary APIs (such as ChatGPT[2]), we finetune an open source model on a generic answer equivalence dataset (Bulian et al., 2022) to score question answering results and evaluate it as a metric on a manually annotated answer equivalence dev set. We call this new metric Gemma Equivalence Metric (GEM).

To summarise, we make the following contributions: (i) We propose a scalable pipeline to generate complex QAD annotations for any video that halves rater time compared to manual annotation. (ii) We use this pipeline to generate the Neptune evaluation-only dataset, which consists of 3,268 QAD annotations for 2,405 videos. We also release a *challenging* subset, NEPTUNE-MMH for which *vision* plays an important role. (iii) We provide both multiple choice and open-ended evaluation metrics. For the latter, we propose a new open-ended metric called Gemma Equivalence Metric (GEM) which outperforms rule-based metrics on a manually annotated answer equivalence dataset; and finally (iv) We provide benchmarking and ablations of state-of-the-art VideoQA models on the Neptune sets. Benchmarking shows a significant gap between open-source video models and proprietary models such as Gemini-1.5 and GPT-4. All data will be released publicly to the research community.

## 2 RELATED WORKS

**Video Question Answering:** Video Question-Answering (VideoQA) is an important task for assessing multimodal video understanding systems' ability to reason about videos (Xu et al., 2017; Zhong et al., 2022; Xiao et al., 2021; Yang et al., 2021; Mangalam et al., 2023). Vision and language models for this task can be broadly classified into three categories: (i) early end-to-end VLMs for this task which typically consists of strong vision and language encoders/decoders, such as Flamingo (Alayrac

---
[1] Named after the planet with the longest orbit
[2] https://openai.com/index/chatgpt/

et al., 2022), BLIP2 (Li et al., 2023b), Video-Llama (Zhang et al., 2023a), GIT2 (Wang et al., 2022) and PALI (Chen et al., 2022; 2023a;b). These typically are moderate sized models, and memory limits often lead to significant downsampling: *e.g.* temporally sampling a few frames with large strides (Wang et al., 2022; Chen et al., 2023a) or spatially subsampling each frame to a single token (Yang et al., 2023; Zhou et al., 2018; Wang et al., 2021); (ii) Socratic style models (Zeng et al., 2022), which consists of combining various specialised *frozen* models with carefully prompted state-of-the-art VLMs and LLMs (eg. MoreVQA (Min et al., 2024)) and (iii) end-to-end large multimodal models such as Gemini (Gemini Team Google, 2023) and GPT-4 (Achiam et al., 2023), which have long context lengths and can ingest multimodal data, including video, sound and text.

**Video QA Benchmarks:** Key datasets have pushed towards assessing reasoning for temporal questions (Grunde-McLaughlin et al., 2021; Xiao et al., 2021; Wu et al., 2021), longer videos (Yu et al., 2019a; Mangalam et al., 2023), as well as focusing on diverse domains like instructional (Yang et al., 2021) and egocentric videos (Gao et al., 2021; Mangalam et al., 2023). We summarise existing VideoQA benchmarks in a table provided in the appendix. Most datasets either focus on shorter videos (less than 100s), or are short video datasets 'in disguise', and can actually be solved with a few frames (*e.g.* ActivityNet-QA (Yu et al., 2019b) or MovieQA (Tapaswi et al., 2016)). 1H-VideoQA (Reid et al., 2024) consists of videos longer than 1 hour, but is limited to 125 questions and is closed-source. Like Neptune, ActivityNet-RTL (Huang et al., 2024), CinePile (Rawal et al., 2024) and EgoSchema (Mangalam et al., 2023) are generated by prompting LLMs, but cover only limited domains and rely on existing annotations while Neptune covers a much broader spectrum of video types and its pipeline is applicable to arbitrary videos. Most importantly, EgoSchema also has strong linguistic biases, while Neptune mitigates these through filtering (Sec. 5). Unlike other benchmarks which come with their own training sets (eg. MSR-VTT (Xu et al., 2016), ActivityNet (Yu et al., 2019a)), we propose a generalisation-focused *zero-shot* evaluation regime. The goal for Neptune is to benchmark any model, pre-trained with any external dataset or task, in order to assess real-world domain transfer. Hence we release *test* sets only. More discussion on related datasets and dataset pipelines is provided in the appendix.

**Metrics for open-ended VideoQA:** Earlier QA datasets consisted of short answers (Xiao et al., 2021) (sometimes a single word), typically from a closed set, and therefore metrics such as accuracy or accuracy with exact match (EM) can be applied. As datasets have evolved with more real-world annotation (longer, open-set answers), designing a metric becomes challenging. Existing rule-based metrics for captioning, such as BLEU (Papineni et al., 2002), ROUGE (Lin, 2004) and CIDEr (Vedantam et al., 2015) can be applied, however they all primarily measure n-gram overlap, and do not capture the inherent subjectivity of the task, where different phrasing is often equally valid. Other metrics for captioning include SPICE (Anderson et al., 2016) (adds action and object relationships), while model-based metrics using earlier language models or image-language models include BERT-Score (Zhang et al., 2020), BERT-Score++ (Yi et al., 2020) (fine-tunes BERT for image captioning), LEIC (Cui et al., 2018), NUBIA (Kane et al., 2020), TIGEr (Jiang et al., 2019), CLIPScore (Hessel et al., 2021), and EMScore (Shi et al., 2022). For answer equivalence specifically, token F1 and exact match (EM) have been used, but suffer many of the same shortcomings that rule-based metrics do, and EM is often too strict for open-ended eval. BEM (Bulian et al., 2022) finetunes BERT on an answer-equivalence dataset, and shows that this provides a better score for QA. Recently, LLMs trained with reinforcement learning from human feedback (RLHF) that already exhibit strong human alignment (Bubeck et al., 2023) are used in works such as VideoChatGPT (Maaz et al., 2023) and MovieChat (Song et al., 2023) (LLM-as-a-judge). A challenge here is that the models used (ChatGPT) are called via proprietary APIs, where the underlying model may be non-static, thereby leading to non-reproducability in the metric. Instead, we take a state-of-the-art open-sourced lightweight language model (Team et al., 2024a) and finetune it on a public answer equivalence dataset (Bulian et al., 2022), to create an open-source, static, model-based evaluation metric.

## 3 NEPTUNE

In this section we describe our dataset generated by the pipeline described in Sec. 4. We first discuss motivating principles, which affect much of the prompt design in the pipeline stage (Sec. 4). Each video contains one or more annotation sets, which consists of a question, an answer to the question and four decoys (which are used for multiple choice evaluation). Our key motivation is that questions should not be answerable by: (i) looking at a single (or few) frames; (ii) using text-only LLMs alone (language, common sense) that have no access to the video; (iii) with only the video's

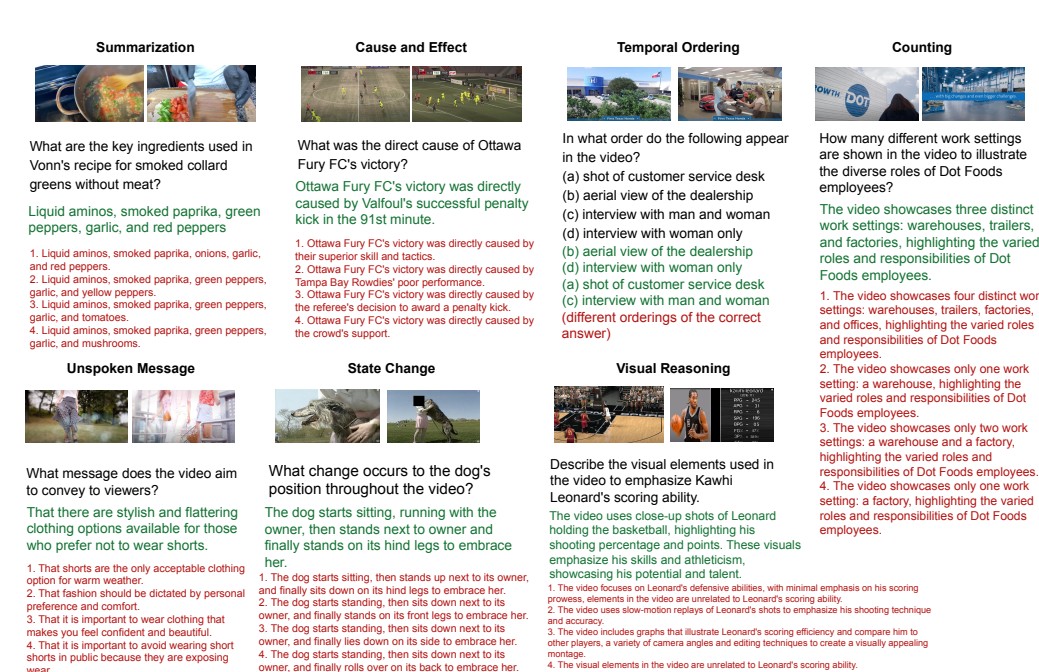

Figure 2: **Examples from Neptune:** We show examples from the dataset that highlight key question types from our dataset. We show 2 frames from each video. Correct answer is provided in green and decoys are shown in red. Best viewed zoomed in and in colour. Some decoys are summarised for brevity.

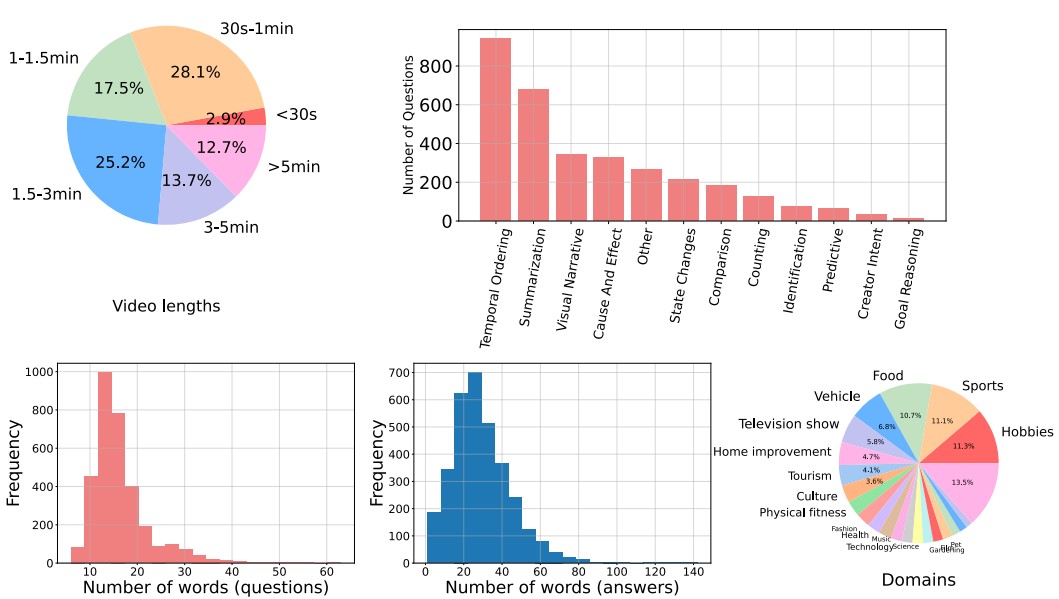

Figure 3: **Neptune Statistics:** We show, the distribution of video lengths (top, left), the number of questions per question type (top, right), the distribution question and answer lengths (bottom, left and middle) and the domains in Neptune (bottom, right). Note that greater than 12% of the videos are longer than 5 minutes (305) and over 25% are longer than 3 minutes. An expanded plot of the video domains is provided in the appendix.

speech transcript, and (iv) questions should cover a number of high-level 'question types', which are discussed next and described in more detail in the appendix.

**Question Types.** Neptune covers a broad range of long video reasoning abilities, which are provided as 'question type' labels for each question. Examples are provided in Fig. 2, and the distribution of questions per question type is depicted in Fig. 3 (right). More information about the distribution of question types is provided in the appendix. Question types are obtained by carefully prompting an LLM (described in Sec. 4.3) and include *Video Summarisation*, which involves summarising and comparing long parts of the video, as well as identifying the most important segments of the video; *Visual Reasoning*, which involves understanding visual elements, as well as reasoning about why visual content is used (*e.g.* to convey a certain mood); *Temporal Ordering*, including the timeline of events; *State Changes*; *Counting* of higher level instances; *Cause and Effect*, and understanding the *Unspoken Message* or *Creator Intent* in a video.

**Dataset Statistics.** Our dataset consists of **3,268** questions from **2,405** videos, covering **100** hours of video. We truncate videos longer than 15 minutes, with the smallest video being 16 seconds and the average length of videos being 2.5 minutes. We show the distribution of video lengths in Fig. 3 (top, left). Note that greater than 12% of the videos are longer than 5 minutes (305 videos) and over 25% are longer than 3 minutes, which is the maximum length of videos in the EgoSchema dataset. The distribution of questions per question type is depicted in Fig. 3 (top, right). The most frequent question type is Temporal Ordering, followed by Summarization. Questions are on average 16.3 words long, while answers and decoys are 29.5 and 29.0 words long respectively. A full distribution of lengths can be seen in Fig. 3 (bottom, left). We also note that the videos in Neptune cover a diverse range of topics (Fig. 3 – bottom, right), an expanded version of this plot is provided in the appendix.

## 4 DATASET CREATION PIPELINE

An overview of our pipeline can be found in Fig. 1. In order to reduce human effort, we leverage automatic tools to (i) find suitable videos (ii) extract useful signals and then (iii) automatically generate video level captions and QADs. We then send the data to human raters for the final manual verification stages. Our pipeline can be applied to any generic YouTube video. This is unlike existing data pipelines such as those used to create EgoSchema (Mangalam et al., 2023), which relies on human generated captions, SFD (Ghermi et al., 2024) and other movie related datasets, which requires movie titles, loglines and synopses (human-written), or MLVU (Zhou et al., 2024), which re-uses annotations from existing datasets for many of their tasks. This makes the dataset scalable, as YouTube has a constantly growing set of videos. Each stage is described in detail below.

### 4.1 VIDEO SELECTION AND SIGNAL EXTRACTION

**Video Selection:** We begin with the YT-Temporal-1Bn (Zellers et al., 2022a) dataset. Because this dataset has strong speech and visual alignment, it consists of a lot of videos where *'talking heads'* dominate the screen (eg. VLOGs, product placements, etc). We attempt to reduce the number of such videos in order to capture more interesting scenes, objects and actions. This is done by extracting face detections with frontal gaze where face bounding-box height is greater than 20%, and removing videos where more than 30% of frames have such frontal gaze. We then apply safety filters to remove racy, local controversy content etc, as well as applying filters to maximise semantic and person diversity. Details about these processes are provided in the appendix.

**Signal Extraction:** For each video we extract the following signals: (i) *Frame captions:* A visual description of each frame (extracted at 1fps) is obtained from PaLI-3 (Chen et al., 2023b). (ii) *ASR:* the speech is transcribed using the YouTube API; (iii) *Metadata:* We obtain the YouTube title and the description for each video; and (iv) *Shot boundaries* for each video.

### 4.2 AUTOMATIC VIDEO CAPTIONING

The signals described above (frame captions, ASR, title and description, shot boundaries) are automatically combined to create video-level captions in a multi-stage process. Examples of caption quality are provided in the appendix, showcasing details such as visual elements, multiple events, mood and atmosphere, details from the ASR, and even high level feelings and emotions. Video captions are obtained using the following steps:

**Shot Visual Captions**: Using the shot boundaries, the *frame captions* are summarized into shot-level descriptions (*shot captions*) by prompting the same LLM. We then create a script for each video containing the shot timestamps, the shot visual captions and the ASR transcript.

**Topic and Description Pairs:** If ASR exists, an initial list of structured topics for the video (along

with a short topic description) is formed by prompting an LLM with the ASR (see appendix). Note that this already yields decent topics as the initial list of videos have been selected (by the YT-Temporal-1Bn authors) to have a strong correlation between ASR and visual content.

**Shot Clustering:** Shots are then clustered per-video using an LLM prompted with the semantic topics obtained above. In each cluster, there may be one or many shots that correspond to that topic. A diagram on this stage and the exact prompt used is provided in the appendix.

**Segment Captions:** Consecutive shots of the same topic are then merged as one segment. Shots of the same topic that are not contiguous are treated as separate segments (see appendix for an example). We then generate dense captions for each segment using a custom prompt (see appendix).

**Adding Visual Support:** To extract a better visual description of the segment that will be used for QA generation in the next phase, an extra step is performed to get visual support for each segment. That visual support is stored separately in conjunction with the dense caption for the segment. For this purpose, the dense caption from the previous step is used alongside the shot level visual captions. The LLM prompt used is provided in the appendix, and the the LLM used for all the above steps is Gemini-1.0-Pro (Gemini Team Google, 2023).

### 4.3 QAD (QUESTION-ANSWER-DECOY) GENERATION

We automatically generate questions, answers and decoys (QADs) by feeding the video captions from above to custom prompted LLMs. Our prompts are inspired by the EgoSchema dataset pipeline (Mangalam et al., 2023), with key modifications to generate more visually focused questions, as well as to generate questions belonging to a set of different question types. The exact prompts used are provided in the appendix. We generate QADs in two stages: (i) Given the video captions from the previous step, we first generate questions and answers; (ii) in the second stage we generate six decoys given the questions and answers from the previous stage. We found this 2-stage method to work better empirically than generating the QADs all in one go.

### 4.4 LLM-BASED BLIND FILTER

**QAD filter:** LLM-based generation can sometimes yield QAD triplets that can be answered from common sense or external world knowledge without the video as context. In particular, we observed that LLMs are often capable of inferring the correct answer from subtle cues in the answer candidates, for example if the correct answer is a positive sentiment while the decoys are negative. To remove such questions, we apply an LLM-based blind filter. We prompt an LLM (Gemini-1.0-pro) to rank the answer candidates to a question. To avoid false rejections due to random correct guesses, we repeat this process three times and only filter out questions where the model predicted the correct answer at least two times out of three (this number was selected to maximise number of videos left given the accuracy trade-off and is discussed in the appendix). Chain-of-thought reasoning improves accuracy so we ask the model to provide a rationale alongside its ranking.

### 4.5 MANUAL RATER VERIFICATION

The final stage involves manual human verification. Raters are first asked to rate the quality of the question based on 4 criteria (details in the appendix). If the question is not suitable, the entire QAD set is discarded. If the question is accepted, raters annotate which modalities are required to answer the question. Choices are: "audio+video", "video-only", or "audio-only". Next, raters are asked to either accept the answer as-is or modify it. Decoys are annotated in a final stage. Given the six LLM-generated decoy candidates, raters verify that they are actually incorrect answers to the question and select the four most challenging ones. If less than four decoys are suitable, we provide a text field for raters to write their own decoys. Screenshots of the rater UI are provided in the appendix. We noticed that rater corrections reintroduce a small amount of questions that can be answered without context, so as a final step we repeat the QAD filter described above. We applied two rounds of manual rater verification to improve dataset quality. More details about rater training, replication (multiple raters per question) and pipelining are provided in the appendix.

**Human Proposed Questions.** To test the effectiveness and efficiency of the automatic portion of our dataset pipeline, we asked raters to propose questions and answers entirely manually for a subset of the dataset. We call this set HPQ (Human Proposed Questions). The raters are provided with a few examples of each question type before they begin annotating. In total, we collect 270 QAs for 193 videos in this set. We use this set in two ways - (i) to quantitatively measure rater-time saved by our automatic pipeline, and (ii) to estimate the amount of Gemini bias in our semi-automatic pipeline. The results for both are provided in Sec. 5.3.

Table 1: Evaluation of open-ended metrics on the GEM answer equivalence dev set. FT: Fine-tuning

| Metric | FT data | F1-Score |
|---|---|---|
| CIDEr (Vedantam et al., 2015) | None | 56.4 |
| ROUGE-L (Lin, 2004) | None | 62.2 |
| BEM (Bulian et al., 2022) | BEM | 61.5 |
| Gemma-2B-IT (Team et al., 2024a) | None | 56.3 |
| Gemma-7B-IT | None | 65.2 |
| Gemma-9B-IT (Team et al., 2024b) | None | 70.3 |
| Gemma-9B-IT (GEM) | BEM | 71.2 |
| Gemini-1.5-pro (Reid et al., 2024) | None | **72.8** |

Table 2: Results on the Human Proposed Question (HPQ) Split. *Results on NEPTUNE-FULL are reported on a subset containing the same set of videos as HPQ.

| Method | Frames | ASR | FULL* | HPQ |
|---|---|---|---|---|
| Video-LLaMA-2 | 16 | No | 13.04 | 14.18 |
| MovieChat | 150 | No | 2.49 | 1.97 |
| MiniGPT4-Video | 45 | No | 5.14 | 4.10 |
| Gemini 1.5 Pro | all | Yes | 45.05 | 44.44 |
| Gemini 1.5 Pro | all | No | 27.67 | 24.81 |

## 5 EXPERIMENTS

We first introduce the two sets in Neptune and our evaluation metrics and then present evaluations using both baseline and state-of-the-art models.

### 5.1 NEPTUNE SETS AND EVALUATION METRICS

**Neptune Sets:** Because we seeded our dataset from the YT-Temporal-1Bn (Zellers et al., 2022b) videos, we note that it contains some videos where ASR can play a big role in contributing to the video content. In order to create a more challenging *visual* benchmark, we also provide Neptune-MMH (multimodal human annotated), where we identify videos where vision should play an important role. This is created by using the rater annotations for what modalities are required to answer the question (described in Sec. 4.5), and discarding questions which the raters marked can be solved by audio-only, and consists of 1,171 QADs for 1,000 videos. We encourage the community to evaluate on this *harder* subset as well.

**Evaluation:** We explore two different protocols for evaluation of question answering - multiple choice evaluation (which involves selecting the correct answer amidst 4 decoys), and open-ended evaluation, which involves producing an answer directly without any decoys and assessing answer quality directly. While the former has the advantage of easier metrics (simple accuracy), the latter removes any potential confounding biases in the decoys. In the next section, we outline our process for creating a new open-ended metric called GEM.

**Gemma Equivalence Metric (GEM):** As discussed in Sec. 2, existing metrics for open-ended QA either lack robustness or rely on proprietary LLM APIs that can change over time. We therefore aim to produce a static open-ended metric. Towards this, we first manually construct a labelled dev-set with 292 (question, reference answer, candidate answer) triplets, with equivalence scores between 0 and 1. See appendix for details on the construction of the dev set. We then benchmark a number of rule-based and model-based metrics on this set in Table 1. To demonstrate the two ends of the scale, we first note that rule-based metrics such as CIDEr (Vedantam et al., 2015) and ROUGE-L (Lin, 2004) obtain F1-Scores of 56.4 and 62.2, while an LLM-based metric using Gemini-1.5-pro (Reid et al., 2024) gets an F1-Score of 72.8 (but is closed-source). Next, we apply static open-source lightweight language models, namely the Gemma family of models i.e. Gemma-2B (Team et al., 2024a), Gemma-7B (Team et al., 2024a) and Gemma-9B (Team et al., 2024b) to judge the answers in a zero-shot setting and find that performance improves with model size, with Gemma-9B bridging the gap well between traditional metrics and the Gemini-1.5-pro based metric. Finally, we fine-tune Gemma-9B on the open-source BEM answer equivalence dataset (Bulian et al., 2022), and find that we obtain a very slight improvement, and hence that it performs the best on our dev-set among the Gemma models. We call the metric obtained with this model Gemma Equivalence Metric (GEM). Note that this metric takes into account the question when comparing whether two answers are equivalent, which is unlike captioning metrics such as CIDEr which omit the question entirely. In Table 4, we report open-ended evaluations using our proposed GEM metric in addition to closed-ended MCQ accuracy. We will release GEM publicly to enable reproducible open-ended evaluations.

### 5.2 BENCHMARKS

We describe all benchmarks used below. Implementation details are provided in the appendix.

**Blind Baselines:** We evaluate models using a text-only prompt in two settings: (i) we feed only the question, answer and decoys to the model (QAD baseline). (ii) we also feed ASR as an input for a QAD+ASR baseline. This helps identify questions that can be answered by prior or commonsense

Table 3: **Ablations using different modalities and number of frames.** † Blind baselines with no access to the video. We show results with one open-source and one closed-source video model.

| Method | ASR | Num. frames | NEPTUNE-FULL Acc. % | GEM | NEPTUNE-MMH Acc. % | GEM |
|---|---|---|---|---|---|---|
| *Open-source* | | | | | | |
| VideoLLaMA2 (Cheng et al., 2024a)† | No | 0 | 38.31 | 4.91 | 30.03 | 0.88 |
| VideoLLaMA2 (Cheng et al., 2024a) | Yes | 0 | **50.15** | **37.50** | 41.23 | **21.83** |
| VideoLLaMA2 (Cheng et al., 2024a) | No | 1 (center) | 40.88 | 16.56 | 36.27 | 14.16 |
| VideoLLaMA2 (Cheng et al., 2024a) | No | 4 | 43.92 | 16.87 | 39.61 | 10.62 |
| VideoLLaMA2 (Cheng et al., 2024a) | No | 8 | 44.74 | 16.26 | 41.32 | 15.93 |
| VideoLLaMA2 (Cheng et al., 2024a) | No | 16 | 44.74 | 17.48 | 40.29 | 15.04 |
| VideoLLaMA2 (Cheng et al., 2024a) | Yes | 16 | 49.28 | 32.54 | **45.38** | 18.18 |
| *Closed-source* | | | | | | |
| Gemini-1.5-pro (Reid et al., 2024)† | No | 0 | 51.53 | 12.12 | 41.84 | 7.59 |
| Gemini-1.5-pro (Reid et al., 2024) | Yes | 0 | 76.68 | **44.92** | 65.76 | 31.20 |
| Gemini-1.5-pro (Reid et al., 2024) | No | 1 (center) | 55.57 | 14.11 | 51.75 | 13.27 |
| Gemini-1.5-pro (Reid et al., 2024) | No | 150 | 69.31 | 25.76 | 66.70 | 22.85 |
| Gemini-1.5-pro (Reid et al., 2024) | No | all | 68.94 | 25.40 | 65.58 | 23.44 |
| Gemini-1.5-pro (Reid et al., 2024) | Yes | all | **80.66** | **44.92** | **75.32** | **34.87** |

knowledge, or ASR only without obtaining visual information from video.

**Image Models:** We use the BLIP2-T5-XL (Li et al., 2023b) model, which contains a 1B vision encoder (Fang et al., 2023) and a 3B text-decoder (Raffel et al., 2020). We feed the center frame of the video as the visual input, with prompt "Answer in one letter" followed by the question and shuffled answer and decoys. We also evaluate some of the video models eg. Gemini-1.5-pro and VideoLLaMA2 as image models, by feeding only the center frame.

**Video Models:** We experiment with 3 different categories of VideoQA models:

(i) Short Context MLLMs - Video-LLaVA (Lin et al., 2023), and VideoLLaMA2 (Cheng et al., 2024b). We also experiment with a simple socratic JCEF (Just Caption Every Frame) (Min et al., 2024), which consists of a VLM to extract per-frame captions and an LLM to perform reasoning on top of these captions to answer the question.

(ii) Long Context MLLMs which are open-source, including MA-LMM (He et al., 2024a), MiniGPT4-Video (Ataallah et al., 2024), and MovieChat (Song et al., 2023).

(iii) Long Context MLLMs which are closed-source, namely the Gemini 1.5 model family (Reid et al., 2024) and GPT-4o (Achiam et al., 2023).

**Implementation Details:** For Video-LLaVA (Lin et al., 2023) we feed 8 uniformly sampled frames (resized to a minimum side length of 320 pixels) along with the question. We reimplement JCEF from the original paper (Min et al., 2024) with updated components - i.e. 16 uniformly sampled frame captions obtained using PaLI-3 (Chen et al., 2023a), and feed them as a text prompt to Gemini-1.0-pro along with the question and decoys. For MiniGPT4-Video, we use the public codebase[3] which routes videos longer than 3 minutes to their Goldfish model and those shorter to their older MiniGPT-video model. We evaluate both the Gemini-1.5-pro and Gemini-1.5-flash models described in (Reid et al., 2024). We also experiment with feeding in ASR to the Gemini-1.5-pro model as well. Frame selection is as other model except that MA-LMM has 20 and 120 and MiniGPT4-Video has default 45 with LLaMA-Video checkpoint. For MA-LMM we feed in 120 uniformly sampled frames. For GPT-4o we use the public API[4].

## 5.3 RESULTS

Results for all the baselines applied to the two Neptune sets (Sec. 5.1) are provided in Table 4. We provide blind baselines and modality ablations in Table 3 for VideoLlaMA2 and Gemini-1.5-pro.

**Single frame baselines:** We examine model performance using the BLIP2 image-only model (Tab. 4) and two video models (VideoLLaMA2 and Gemini-1.5-pro) with only the center frame of the video in Tab. 3. The larger Gemini model outperforms BLIP-2, however performance with only a single frame is much lower than with multiple frames, as expected. We also show results using Gemini-1.5-pro on the first frame of the video in Fig. 4 (right), and find that using the middle frame performs better. VideoLlaMA2 is a short context model, and we find performance saturates at 8 frames. Surprisingly, the best result of VideoLlaMA2 is obtained using ASR only and not providing image frames. In

---

[3]https://github.com/Vision-CAIR/MiniGPT4-video
[4]accessed Sept 30th, 2024

Table 4: **Benchmarking performance on Neptune. All frames:** Visual frames extracted at 1fps. *Computed on 10% of the results. ‡ MCQ performance is close to random.

| Method | Modalities | NEPTUNE-FULL | | NEPTUNE-MMH | |
|---|---|---|---|---|---|
| | | Acc. % | GEM | Acc. % | GEM |
| Random | - | 20.00 | | 20.00 | |
| *Image models* | | | | | |
| BLIP2 (Li et al., 2023b) | RGB (center frame) | 34.80 | 9.20 | 28.10 | 8.50 |
| *Short Context MLLMs* | | | | | |
| Video-LLaVA (Lin et al., 2023) | RGB (8 frames) | 25.79 | 10.66 | 24.00 | 5.48 |
| VideoLLaMA2 (Cheng et al., 2024a) | RGB (16 frames) | 44.74 | 17.48 | 40.29 | 15.04 |
| VideoLLaMA2 (Cheng et al., 2024a) | RGB (16 frames) + ASR | 49.28 | 32.54 | 45.38 | 18.18 |
| *Long Context MLLMs - open-source* | | | | | |
| MA-LMM (He et al., 2024a) (ActivityNet-QA fine-tuned) | RGB (120 frames)‡ | 20.22 | 10.67 | 19.51 | 5.04 |
| MiniGPT4-Video (Ataallah et al., 2024) | RGB (45 frames)‡ | 24.63 | 5.26 | 22.89 | 6.19 |
| MovieChat (Song et al., 2023) | RGB (150 frames) | 28.96 | 3.79 | 30.30 | 1.01 |
| *Closed-source MLLMs* | | | | | |
| VLM captions + LLM (JCEF) (Min et al., 2024) | VLM captions (16 frames) | 58.51 | 12.27 | 56.45 | 11.50 |
| GPT-4o (Achiam et al., 2023) | RGB (8 frames) + ASR | 80.23 | *49.01 | 72.86 | |
| Gemini-1.5-pro (Reid et al., 2024) | RGB (all frames) + ASR | **80.66** | 44.92 | **75.32** | **34.87** |
| Gemini-1.5-flash (Reid et al., 2024) | RGB (all frames) + ASR | 76.90 | 45.59 | 71.05 | 33.93 |

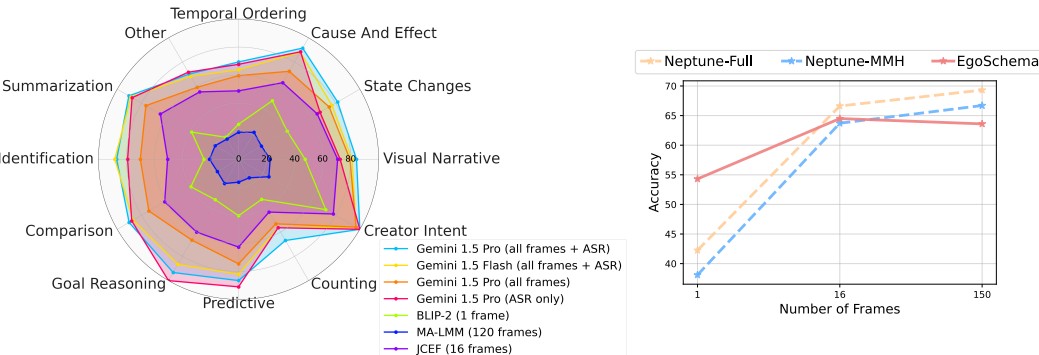

Figure 4: **Performance of different models across question types on NEPTUNE-FULL (left) and Neptune Vs Egoschema with different frame rates (right).** On the right we show Gemini 1.5 Pro's accuracy when linearly subsampling to 1, 16 or 150 frames. We note that (i) performance on the Neptune sets increases as more frames are provided while on EgoSchema it saturates after 16 frames and (ii) NEPTUNE-MMH is more challenging than EgoSchema.

fact, if we provide 16 frames in addition to ASR (last row of the open-source block), performance drops slightly. We assume that this is a result of attention dilution (Coleman et al., 2023), where an increasingly large context distracts the model, causing a drop in performance.

**Modality Ablations:** Table 3 shows that performance of Gemini-1.5-pro with ASR only as input is higher than performance with multiple video frames on the NEPTUNE-FULL set, but not on the NEPTUNE-MMH set. On both sets however, the best performance is obtained with both frames and ASR, showcasing the complementary nature of the modalities.

**Multi-frame reasoning:** We note that Gemini-1.5-pro with multiple frames outperforms the JCEF baseline in Tab. 4, even though the JCEF baseline uses the weaker Gemini-1.0 model (albeit applied to only a single frame at a time). This shows that frames must be processed together (by a video model), and a socratic baseline that simply looks at each frame individually performs much worse on this benchmark. This is unlike other datasets such as Next-QA where JCEF style baselines are almost state-of-the-art (Min et al., 2024).

**Video Models:** We see a significant gap between open-source models and Gemini-1.5-pro and GPT-4o. Interestingly, we find that open-source models that are designed specially for longer context video understanding (MA-LMM (He et al., 2024a), MiniGPT4-Video (Ataallah et al., 2024) and MovieChat (Song et al., 2023)) perform worse than VideoLLaMA2. This observation was also found by concurrent datasets such as MLVU (Zhou et al., 2024) and LVBench (Wang et al., 2024). The gap between many open-source and proprietary large MLLMs is also shown on concurrent datasets, *e.g.* LVBench (Wang et al., 2024), where MovieChat gets near-random results and Gemini-1.5-pro is the state-of-the-art. One reason for this near random performance may be the domain gap between the

training sets of these models (He et al., 2024a; Song et al., 2023) and Neptune – MovieChat is trained on movies and MA-LMM is designed to be fine-tuned on downstream QA datasets. By not providing a training set, we intentionally aim to assess generalization via zero-shot performance. We also note that the simple JCEF baseline, which consists of frame captions fed to an LLM for reasoning, outperforms all open-source models. The low performance of open-source models suggests Neptune may be a challenging benchmark for the future development of open-source models for long videos.

**Challenging split and Gemini Bias:** Both GPT-4o and Gemini-1.5-pro perform comparably on NEPTUNE-FULL, despite Gemini-1.5-pro being used in dataset creation, and on the NEPTUNE-MMH set, neither model is able to achieve saturated performance. This suggests that our extensive human rater step was able to help mitigate Gemini bias. This is unlike VideoVista (Li et al., 2024) which uses GPT-4 to generate QADs automatically. However the performance of GPT-4 and Gemini-1.5 on their dataset is close to saturated (98% on some categories). We note that performance falls for all models universally on the NEPTUNE-MMH set demonstrating the challenging nature of this set.

**Results on HPQ and Gemini bias:** In Tab. 2, we compare open-ended question answering performance on questions generated by our pipeline to performance on fully human written questions (HPQ) on the same set of videos. The time taken to manually create HPQ (19.03 minutes on average per question) is significantly longer than simply discarding or correcting QAs generated automatically as is done in our Neptune pipeline (10.32 minutes). While most models perform slightly worse on HPQ, overall performance is similar, suggesting that our automatic pipeline reaches the same difficulty level roughly half the rater effort. Notably, Gemini-1.5-pro performs comparatively on both sets, suggesting that bias introduced by the model in the creation pipeline is limited.

**Video Coverage compared to EgoSchema:** In this section we investigate Gemini 1.5 Pro's accuracy when linearly subsampling the video to 1, 16, or 150 frames. For 1 frame, we take the first frame of the video. We show results for all Neptune splits and compare them to results on EgoSchema in Fig. 4. Gemini 1.5 Pro's performance on Neptune increases as more frames are provided, while on EgoSchema it saturates after 16 frames, suggesting Neptune is better at requiring *long* video reasoning. Note that every video in EgoSchema has 180 frames (3 mins), whereas Neptune has variable lengths, with videos up to 15 minutes long. Results with the first frame on both Neptune splits are also much lower than those on EgoSchema (54.3), pointing to higher image bias in the latter. EgoSchema also introduced the concept of a temporal certificate. We introduce a slightly modified version, which is *Model-Based*, and show that the Gemini-1.5-pro model needs more frames to answer a question correctly in Neptune, with a mean certificate of 5.39 frames (compared to 1.6 for EgoSchema). The details of this experiment are provided in the appendix.

**Open-ended results:** We find that in general, results with GEM mirror the trends demonstrated by the multiple choice eval, with the exception of the Gemini-1.5-flash and Gemini-1.5-pro results, as well as the performance of the long context open-source models. Here we find that the FLASH model actually slightly exceeds the performance of the PRO model on the FULL set, and MovieChat performs worse on the open-ended task than other baselines, while better on the MCQ evaluation. A qualitative examination of the scores with the highest disparity shows that the FLASH model seems to indeed provide better open-ended answers. Examples of this are provided in the appendix.

**Results per question type:** Performance of different models across the different question types are shown in Fig. 4. We find that "Counting", "Temporal Ordering" and "State Change" questions are challenging for all models, pointing to areas for future work for video-language models, while "Cause and Effect" is easier. Interestingly, the Gemini-1.5-Pro model applied only to ASR without access to video frames is the best at "Goal Reasoning", which may be because human goals in videos are often mentioned in speech. Yet as expected, it is worse at the "Visual Narrative" questions, where Gemini-1.5-Pro models with access to RGB frames do much better.

## 6 CONCLUSION

We present Neptune, a new benchmark for VideoQA with a focus on *multimodal*, *high-level* understanding of *long videos*. Neptune is created using a scalable pipeline for arbitrary videos that minimizes (though not omits) human verification. Benchmarks are evaluated using MCQ and open-ended evals – for which we provide a new, open-source metric. **Limitations:** The dataset may inherit biases of the Gemini model used to generate QADs. While VideoQA is a good proxy for video understanding, our dataset could be further improved by additional annotations – such as manually annotated temporal grounding, dense captions or entity labels.

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
