# 7 APPENDIX

## A RELATED WORKS

Here we provide an additional discussion of related works that were omitted from the main paper due to lack of space. The recently released Perception Test (Patraucean et al., 2024) consists of script-based recorded videos with manual annotations focusing on 4 broad skill areas - Memory, Abstraction, Physics, Semantics, however videos are only 23s long (avg). Like Neptune, ActivityNet-RTL (Huang et al., 2024) was constructed in a semi-automatic fashion by querying GPT-4 to generate comparative temporal localization questions from the captions in ActivityNet-Captions (Krishna et al., 2017). CinePile (Rawal et al., 2024) was generated by prompting an LLM to generate multiple-choice questions. Because it is based on movie clips, it can leverage available human-generated audio descriptions. Both ActivityNet-RTL and CinePile cover only limited domains and rely on existing annotations while Neptune covers a much broader spectrum of video types and its pipeline is applicable to arbitrary videos. Our rater stage is lightweight, unlike other works that are entirely manual (Zhou et al., 2024; Fang et al., 2024; Wang et al., 2024). In LVBench (Wang et al., 2024), even the video selection is done manually, and for MoVQA (Zhang et al., 2023b), only the decoys are generated automatically. Another recently released dataset (concurrent with our submission) is the Video-MME dataset (Fu et al., 2024). The motivation of this dataset is similar to ours, namely it covers videos of variable lengths, with 2,700 QADs covering a wide range of different question types. The main difference between Video-MME and Neptune is that the former is entirely manually annotated by the authors, while we propose a scalable pipeline which can be applied to new videos and domains automatically, and can be tweaked to include different question types with reduced manual effort. EgoSchema is the closest work to ours in motivation, but there are some key differences: (i) it is limited to egocentric videos of exactly 3 minutes each, while Neptune covers many domains and follows a more natural length distribution for online videos (16s to 15min); (ii) it relies heavily on manually obtained dense captions for egocentric videos, while our method generates captions automatically too and hence can be easily applied to any video online; and more importantly (iii) EgoSchema also has strong image and linguistic biases, while Neptune mitigates these.

Table 5: **Comparison to Existing VideoQA datasets: Ann. Type:** Annotation Type, **QAD:** Question, Answer and Decoys, **Rater V:** Rater verified manually. † Movies are no longer available. ‡ Annotations are hidden behind a test server, 500 are public. *average/max length. **short/medium/long.

| Name | Ann | Rater V | Avg. len (s) | # Vids (total/test) | # Samples (total/test) | Available |
|---|---|---|---|---|---|---|
| MovieQA (Tapaswi et al., 2016) | QAD | ✓ | 200 | 6,771/1,288 | 6,462/1,258 | ✗† |
| MSRVTT-QA (Xu et al., 2017) | QA | ✗ | 15 | 10,000/2,990 | 243,680/72,821 | ✓ |
| ActivityNet-QA (Yu et al., 2019a) | QA | ✓ | 180 | 5,800/1,800 | 58,000/18,000 | ✓ |
| NExTQA (Xiao et al., 2021) | QAD | ✓ | 44 | 5,440/1,000 | 52,044/8,564 | ✓ |
| IntentQA (Li et al., 2023a) | QAD | ✓ | 44 | 4,303/430 | 16,297/2,134 | ✓ |
| EgoSchema (Mangalam et al., 2023) | QAD | ✓ | 180 | 5,063/5,063 | 5,063/5,063 | ✓‡ |
| Perception Test (Patraucean et al., 2024) | QAD | ✓ | 23 | 11,600 | 38,000 | ✓ |
| MVBench (Li et al., 2023c) | QAD | ✗ | 16 | 3,641 | 4,000 | ✓ |
| Video-Bench (Ning et al., 2023) | QAD | ✓ | 56 | 5,917 | 17,036 | ✓ |
| AutoEval-Video (Chen et al., 2023c) | QA | ✓ | 14.6 | 327 | 327 | ✓ |
| 1H-VideoQA (Reid et al., 2024) | QAD | ✓ | 6,300 (max) | 125 | 125 | ✗ |
| MLVU (Zhou et al., 2024) | QAD | ✓ | 720 | 2K | 2593 | ✓ |
| Video-MME Fu et al. (2024) | QAD | ✓ | 82.5/562.7/2,385.5** | 900 | 2,700 | ✓ |
| LongVideoBench Wu et al. (2024) | QAD | ✓ | 473 | 3,763 | 6,678 | ✓ |
| **Neptune** | QAD | ✓ | 150/901* | 2,405 | 3,268 | ✓ |
| **Neptune-MMH** | QAD | ✓ | 159/901* | 1,000 | 1,171 | ✓ |

## B THE NEPTUNE DATASET

### B.1 ADDITIONAL INFORMATION ON QUESTION TYPES

Neptune covers a broad range of long video reasoning abilities, which are summarised below. These question types are obtained in the Question and Answer generation stage, for which the prompt is provided in Sec. C.2.3. We provide further insights into the motivations of some of the question areas provided in the prompt below.

**Video Summarisation:** Summarise and compare long parts of the video, as well as identify the most

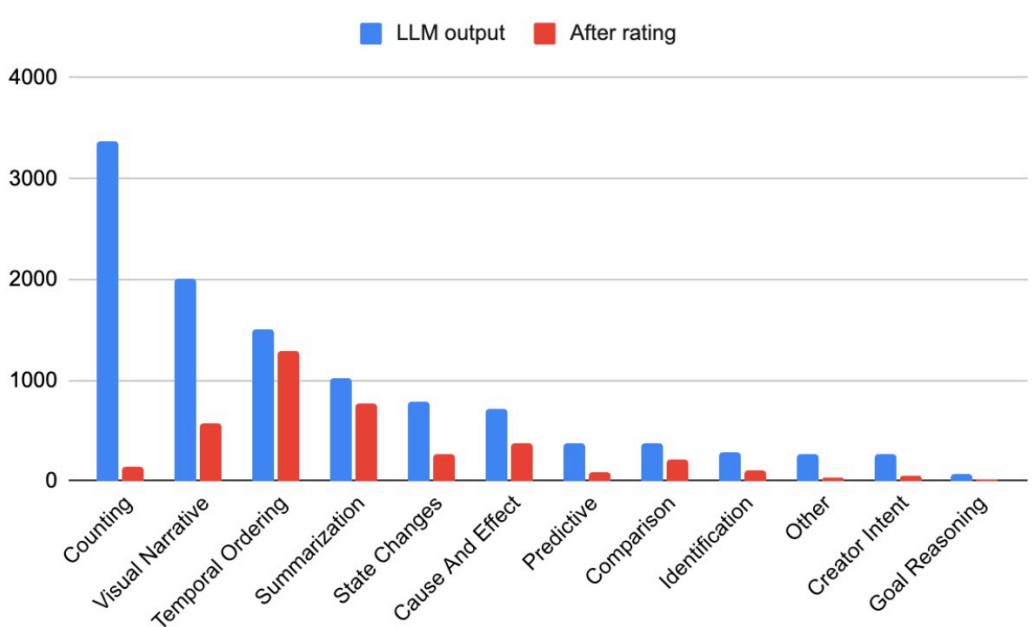

Figure 5: Change of question type distribution as a result of human rater filtering.

important segments of the video.

**Visual Reasoning:** Recognize and understand visual elements in different parts of the video, as well as reason about why visual content is used (*e.g.* to convey a certain mood).

**Temporal Ordering:** Understand the timeline of events and the plot in the video.

**Counting:** Count objects, actions and events. Here we focus on higher-level counting where the same instance does not occur in all/every frame and actions are sufficiently dissimilar.

**Cause and Effect:** Understand and reason about cause and effect in the video.

**Message:** Understand the unspoken message that the audience may perceive after watching the video, which may require common sense knowledge to infer.

**State Changes:** Understand object states change over time, such as a door opening and food being eaten.

Since the questions are proposed automatically by an LLM, the question types are also generated in an open-set manner by the LLM. Hence sometimes, the LLM will generate the question type label using different phrasing - eg. 'temporal ordering' or 'timeline event'. We use simple manual postprocessing to group similar question types into the same category, with a few question types that do not fall into any of the categories grouped as 'Other'. The final *question types* released with the dataset are shown in Fig. 3 of the main paper.

### B.1.1 QUESTION TYPE DISTRIBUTION

We explain the reasons for Neptune's current question type distribution:

(i) We prompted the LLM that generated the questions with a set of examples of different question types and let the model choose which questions to generate.

(ii) The model's selection of question types depends strongly on the given video. For example, while it is always possible to ask for a video summary, it is not always possible to ask about a person's goals, or cause and effect, because not all videos allow for these types of reasoning. This naturally leads to an imbalance of possible question types.

(iii) Additionally, we observed that the quality of questions produced by the LLM varies strongly by question type. Therefore, after quality checking by raters, the distribution changes significantly (Fig. 5). The strongest difference was for counting questions, as LLM-proposed questions were often too easy, e.g. counting the number of times a certain word is mentioned.

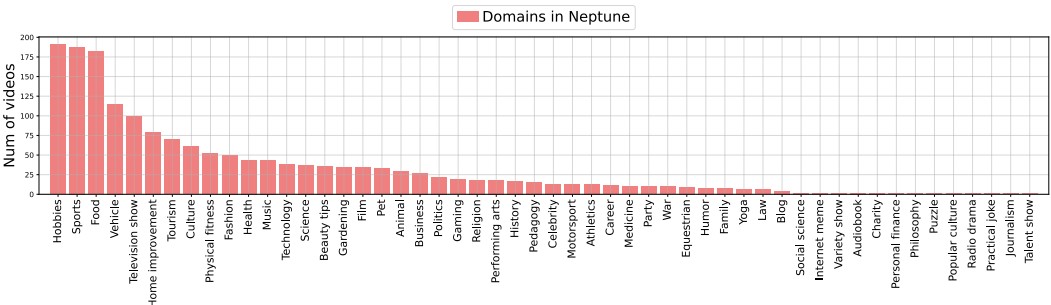

Figure 6: **Domains in Neptune:** We show the number of videos per domain category in NEPTUNE-FULL.

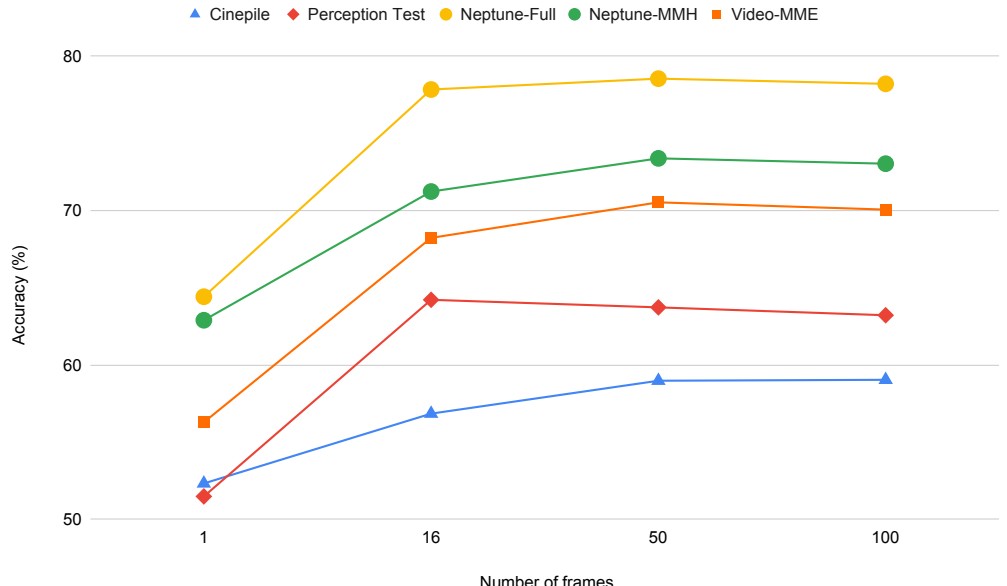

Figure 7: Comparison of Neptune to other video benchmarks. We evaluate Gemini-1.5-Flash with different numbers of frames that are uniformly sampled across the videos.

## B.2 DOMAINS IN NEPTUNE

A full graph of the domains in Neptune are provided in Fig. 6.

## B.3 COMPARISON TO OTHER BENCHMARKS

We measure the complexity of Neptune compared to other benchmarks by analyzing the progression of model performance as we add more frames to the context. We use Gemini-1.5-Flash for this comparison since it is capable of handling very large contexts. Fig. 7 shows the results of this experiment, comparing Neptune with CinePile (Rawal et al., 2024), Perception Test (Patraucean et al., 2024) and Video-MME (Fu et al., 2024). We find that most benchmarks saturate at about 50 frames, including Video-MME, which has much longer videos than Neptune. While we included Perception Test here as it is new, it does not claim to be a long video benchmark and saturates at 16 frames.

## B.4 PER-TASK PERFORMANCE

We provide detailed per-task model performance in Tab. 6. See Fig. 4 (bottom left) for a graphical representation of a subset of these results. Overall, closed-source MLLMs perform best across all

Table 6: Per-task model performance. Tasks are abbreviated as follows: TO: Temporal Ordering, CE: Cause And Effect, SC: State Changes, VN: Visual Narrative, CI: Creator Intent, CT: Counting, PR: Predictive, GR: Goal Reasoning, CMP: Comparison, ID: Identification, SUM: Summarization, OTH: Other. The best accuracy per task is printed in bold and the second best underlined.

| Method | Modalities | TO | CE | SC | VN | CI | CT | PR | GR | CMP | ID | SUM | OTH | Task-avg |
|---|---|---|---|---|---|---|---|---|---|---|---|---|---|---|
| *Image models* | | | | | | | | | | | | | | |
| BLIP2 (Li et al., 2023b) | RGB (center frame) | 24.97 | 48.18 | 40.09 | 47.51 | 71.88 | 33.06 | 40.30 | 33.33 | 39.34 | 24.68 | 38.64 | 18.11 | 38.34 |
| *Short Context MLLMs* | | | | | | | | | | | | | | |
| Video-LLaVA (Lin et al., 2023) | RGB (8 frames) | 22.95 | 36.06 | 28.30 | 30.79 | 46.88 | 19.35 | 35.82 | 53.33 | 31.15 | 20.78 | 20.06 | 23.40 | 30.74 |
| VideoLLaMA2 (Cheng et al., 2024a) | RGB (16 frames) | 35.71 | 57.27 | 48.36 | 57.31 | 78.13 | 33.06 | 53.73 | 60.00 | 50.54 | 42.86 | 47.35 | 29.81 | 49.51 |
| VideoLLaMA2 (Cheng et al., 2024a) | RGB (16 frames) + ASR | 34.08 | 60.00 | 53.77 | 59.06 | 90.63 | 38.71 | 56.72 | 73.33 | 61.96 | 54.55 | 59.00 | 35.09 | 56.41 |
| *Long Context MLLMs - open-source* | | | | | | | | | | | | | | |
| MA-LMM (He et al., 2024a) | RGB (120 frames) | 19.34 | 22.12 | 18.87 | 22.58 | 25.00 | 15.32 | 16.42 | 20.00 | 17.49 | 20.78 | 19.32 | 16.60 | 19.49 |
| MiniGPT4-Video (Ataallah et al., 2024) | RGB (45 frames) | 20.43 | 34.24 | 24.06 | 30.79 | 34.38 | 21.77 | 31.34 | 33.33 | 21.31 | 32.47 | 23.16 | 21.89 | 27.43 |
| LLaVA-OneVision (Li et al., 2024a) | RGB (100 frames) | 57.83 | 73.33 | 72.99 | 77.71 | 84.38 | 41.94 | 82.09 | 86.67 | 67.21 | 71.43 | 71.39 | 55.47 | 70.20 |
| MiniCPM-V 2.6 (Yao et al., 2024) | RGB (50 frames) | 41.32 | 65.15 | 67.3 | 70.38 | 75.0 | 37.9 | 67.16 | 86.67 | 60.66 | 66.23 | 66.22 | 46.42 | 62.53 |
| *Closed-source MLLMs* | | | | | | | | | | | | | | |
| JCEF (Min et al., 2024) | VLM captions (16 frames) | 48.78 | 63.03 | 64.79 | 70.76 | 78.13 | 43.55 | 62.69 | 60.00 | 60.87 | 50.65 | 64.45 | 55.47 | 60.26 |
| GPT-4o[4] (Achiam et al., 2023) | RGB (8 frames) + ASR | **71.25** | 91.21 | 77.25 | 76.83 | **100.0** | 62.90 | **89.55** | 93.33 | 87.98 | 85.71 | **91.30** | **72.45** | 83.31 |
| Gemini-1.5-pro (Reid et al., 2024) | RGB (all frames) + ASR | 69.39 | **91.52** | **81.69** | **84.21** | **100.0** | **66.94** | 86.57 | 93.33 | **90.22** | 87.01 | 90.41 | 70.19 | **84.29** |
| Gemini-1.5-flash (Reid et al., 2024) | RGB (all frames) + ASR | 63.87 | 88.18 | 77.00 | 80.99 | 96.88 | 56.45 | 82.09 | 86.67 | 86.96 | **88.31** | 88.79 | 68.30 | 80.37 |

tasks, with Gemini-1.5-pro ranking best overall and GPT-4o ranking second. Even though their average scores are close, there are significant differences in per-task scores, showing the different capabilities of each model.

# C    IMPLEMENTATION DETAILS

## C.1    VIDEO SELECTION

We choose the YT-Temporal-1Bn dataset (Zellers et al., 2022b) as the source for Neptune, because of its large and diverse corpus, and because of the high correlation between vision and audio transcripts.

**Safety & Content Filters:** We filter out videos with less than 100 views, that are uploaded within 90 days, and those tagged by YouTube content filters to contain racy, mature or locally controversial content. We then identify and remove static videos (eg. those that consist of a single frame with a voiceover) by clustering similar frames in a video and ensure that there is more than 1 cluster. We also identify and remove videos comprising primarily of "talking heads". To achieve this, we apply a per-frame frontal-gazing face-detector at 1fps and mark the frames where the bounding box height is greater than 20% as *talking head frames*. Then, we filter out videos where more than 30% of the frames are talking head frames. These thresholds are chosen based on an F1-score on a small dev set of 50 manually annotated videos.

**Diversity Sampling:** From the filtered set of videos, we sub-sample $100,000$ videos to boost both semantic and demographic diversity. First, we cluster the videos based on video-level semantic embeddings and tag each video with a cluster id. Second, we tag each video with the perceived age and gender demographic information contained in the video. Third, we obtain a joint distribution of semantics (cluster id) and demographics (perceived age and gender) and apply a diversity boost function (Kim et al., 2022) on the joint distribution. Finally, we sample from videos from this distribution. Fig. 8, shows the down-sampling of over-represented cluster ids before and after applying the filter. We then uniformly sub-sample the videos further to reach the desired dataset size.

## C.2    PROMPTS FOR DATA GENERATION

In this section we provide some of the prompts used for generating Neptune.

### C.2.1    PROMPT FOR FRAME CAPTIONING

We use the following prompt to obtain a caption for each video frame:

```
Answer the following questions about the given image.  Then use the
information from the answers only, and write a single sentence as caption.
Make sure you do not hallucinate information.

Question(Mood):  Describe the general mood in the image as succinctly as
possible.  Avoid specifying detailed objects, colors or text.
```

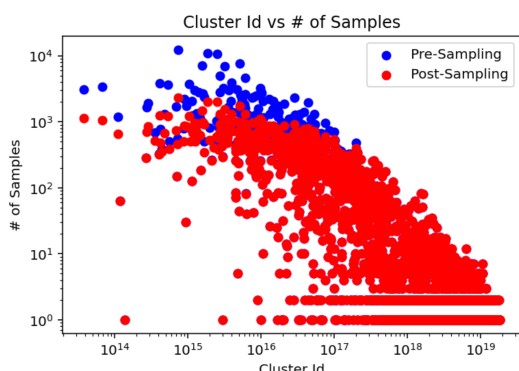

Figure 8: **Diversity sampling:** We show the change in cluster distribution after diversity sampling.

```
Question(Background):  Describe the background of the image as succinctly
as possible.  Avoid specifying detailed objects, colors or text.  Eg:  The
background is a parking lot, playground, kitchen etc.
Question(Person):  Is there any person in the image.  If yes, describe them
and what are they doing here?  If no, say no person.
Question(General):  Describe the image as succinctly as possible.  Avoid
specifying detailed objects, colors or text.
Question(Text):  Is there any text?  What does it say?
```

```
Result template:
```

```
Answer(Mood):  A succinct description of what is happening in the image
with the general mood.
Answer(Background):  A succinct description of the background scene in the
image and what is happening.
Answer(Person):  If there are people in the image, a succinct description.
Answer(General):  A succinct description of the image.
Answer(Text):  Reply if there is any text, where it is placed and how it is
related to what is happening in the image.
```

```
Caption:  A couple of sentences summarizing the information given by the
answers about mood, background, person, general and text.
```

```
With the above format as template, generate the response for the new image
next.
```

### C.2.2 PROMPTS FOR AUTOMATIC VIDEO CAPTIONING

A visual overview of the video captioning stage is provided in Fig. 9. We describe the prompts for each stage below:

**Shot level captions:**
Using the shot boundaries the 1fps frame captions are summarized into shot level descriptions with the following prompt:
```
Summarize these sentences in dense short sentences:  [list of frame
captions in the shot]
```

**Topic and Description Pairs:**
If ASR exists, topic and description pairs are obtained from ASR using the following prompt:
```
**Task:** Take a deep breath and give me the structural topics of the
Youtube video below using the transcript.  Give up to 5 Topic and
Description pairs using output format.  **Transcript:** transcript
```

**Shot Clustering:**
```
Take a deep breath and identify the sequential topic structure of this
video using the "{head_topic}" in Scenes.  A part of the video script
```

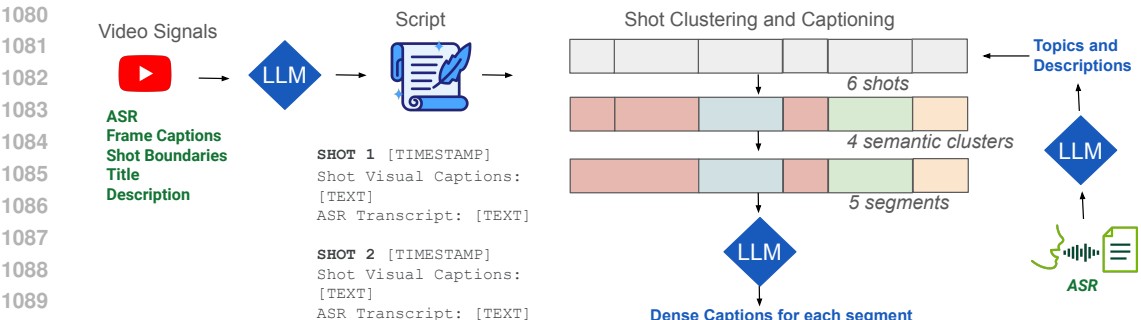

Figure 9: **Video Captioning:** We extract dense segment level captions automatically for each video. This is done by prompting an LLM using video signals (ASR, frame captions, shot boundaries and metadata) with various different steps and prompts.

```
is provided as a set of Scenes and in each scene, visual captions and
transcript sentences are provided.  The overall suggested structure from
the transcript is provided as well.  Assign every scene in this part of
the script to one topic structure.  For each scene, the visual captions
should support and relate the topic.  If the support or relation is not
strong create a new topic and assign the scene to it.  Reevaluate the
suggested structure from the transcript and make sure all scenes are
assigned to the best associated topics.  Keep output length to be less than
{max_output_characters} characters.
```

```
**Output Format:** XML output where topic has the following children
(description, topic_scenes, story) <topic> <description>The description
of the topic</description> <topic_scenes>Comma separated scene number(s)
related to this topic<topic_scenes> <story>Summarized caption that
describes what happens and what's shown for this topic in the scenes
by combining visual caption and transcript sentences of the related
scenes</story> </topic>
```

```
**Suggested Structure:** {initial_structure_from_ASR_if_exists}
```

```
**Context:** {summary_of_title_and_description}
```

```
**Video Script:** {video_script}
```

**Segment Captions:**

Consecutive shots of the same topic are then merged as one segment. Shots of the same topic that are not contiguous are treated as separate segments (see Fig. 9). We then generate dense captions for each segment using the prompt below:

```
**Task:** You are the expert in video description writing.  Use the
information "Partial Script" to improve the "Initial Description" by
adding the missing information either from visual or transcript.  The
video context is also given to help you interpret the script.  Only add
information that is in the "Partial Script".  Make the output concise
and compact with less or the same length as Initial Description.  The
updated video description is plain text.  Your answer should follow the
output format.  Keep output length to be less than max_output_characters
characters.
```

```
**Initial Description:*** shared_topic_cluster_caption
```

```
**Output Format:** XML format like below <updated_description>updated video
description text</updated_description>
```

```
**Partial Script:** doc_segment
```

**Visual Support Caption**
To extract better visual description of the segment that will be used for QA generation in the next phase, an extra step is performed to get visual support for each segment. That visual support is stored separately in conjunction with the dense caption for the segment. For this purpose, the dense caption from the previous step is used alongside the shot level visual captions. The following LLM prompt is used to extract the visual support:

```
**Task:** I provide video scene information and your job is to summarize
the exact elements from "Visual Captions" that directly support the "Scene
Story" of the scene below.  The visuals of the scene is broken down to
shots and each shot is described in a line of text in the Visual Captions.

**Scene Story:** dense_caption_for_the_segment

**Visual Captions:** visual_captions_of_the_segment

**Output Format:** Plain text with at most 200 words summarizing the
supporting visual elements.
```

C.2.3 GENERATING QUESTIONS AND ANSWERS

```
I want you to act as a rigorous teacher in the "Long-term Video
Understanding" class.  Let's test your students' in-depth comprehension!

Understanding:  I'll provide you with the following:

- Dense Captions:  A detailed breakdown of the video, including key moments
and timestamps.  Analyze this carefully.

Your Task:  Craft {target_number} Challenging Short-Answer Questions

Requirement:

- Challenge:  Demonstrate your ability to create challenging, insightful
short-answer questions about the video.  These shouldn't test simple recall
only.  Aim to probe understanding of relationships, motives, subtle details,
and the implications of events within the video.
- Diversity:  Design a variety of question types (more on this below).
- Specificity:  Each question must be self-contained and laser-focused on
a single concept or event from the video.  Avoid compound or overly broad
questions.
- Answers:  Model the ideal answer format:  Brief, accurate, and rooted
directly in evidence from the video's content.
- Video-Centric:  Stay true to what's explicitly shown or stated in
the video.  Avoid relying on outside knowledge or speculation.  Design
questions so the correct answer cannot be easily determined without
carefully analyzing the video.
- Minimize Information Leakage:  For question types like ranking or
ordering, ensure that the order of candidates or options listed in the
question doesn't inadvertently reveal the correct answer.  Shuffle them to
maintain neutrality.
- Content-First:  Timestamps and section titles within the captions are
there for guidance.  Do not explicitly refer to those markers in your
questions or answers.  Focus on the events and elements themselves.
- Unambiguous:  Ensure each question has a single, clearly defined correct
answer.  Avoid questions that are open to multiple interpretations (e.g.,
counting elements where viewers might disagree).
- Visual Elements:  Questions focused on visual reasoning or visual
narratives should emphasize the interpretation of the visuals.  Keep the
question minimal, letting the answer describe the specific visual elements
in detail.
```

You want to test students' capabilities of understanding the video, including but not limited to the following aspects:

Ability:  Summarize and compare long parts of the video.
Ability:  Compress information from the video rather than just listing the actions that happened in the video.
Ability:  Identify the most important segments of the video.
Ability:  Recognize and understand the visual elements in different parts of the video.
Ability:  Understand the timeline of events and the plot in the video.
Ability:  Count objects, actions and events.  Focus on higher-level counting where the same instance does not occur in all/every frame and actions are sufficiently dissimilar.
Ability:  Understand and reason about cause and effect in the video.
Ability:  Understand the unspoken message that the audience may perceive after watching the video, which may require common sense knowledge to infer.
Ability:  Understand the visual reasoning of why and how important visual content is shown in the video.
Ability:  Understand the visual narrative of the video and the mood of the video and which visual elements do contribute to that.
Ability:  Understand object states change over time, such as door opening and food being eaten.

Presentation

- QUESTION: Introduce each question as "QUESTION 1, 2, 3:  (capability) full question".  - ANSWER: Follow the format "CORRECT ANSWER: correct answer".

Good example questions:  - Question (counting):  How many ingredients are added to the bowl in total throughout the video?  Correct Answer:  3.

- Question (goal reasoning):  What is the purpose of the man standing in front of the whiteboard with a diagram on it?  Correct Answer:  To explain the features and capabilities of the vehicle.

- Question (cause and effect):  How does the document help people to be happier?  Correct Answer:  It helps people to identify and focus on the things that make them happy, and to develop healthy habits.

- Question (timeline event):  In what order are the following topics discussed in the video:  history of pantomime, importance of pantomime, mime as a tool for communication, benefits of pantomime?  Correct Answer:  Mime as a tool for communication, history of pantomime, importance of pantomime, benefits of pantomime.

- Question (predictive):  What happens after the man jumps up and down on the diving board?  Correct Answer:  He jumps into the pool.

- Question (summarization):  What is the overall opinion of the reviewers about Hawaiian Shaka Burger?  Correct Answer:  The food is good, but the patties are frozen.

- Question (creator intent):  What message does the video creators try to send to the viewers?  Correct Answer:  Nature is essential for human well-being.

- Question (visual-temporal):  What color is the scarf that Jessica wears before she enters the restaurant?  Correct Answer:  Red.

- Question (visual narrative):  How does John's overall facial expression contribute to the explanation of the financial situation that is described in the video?  Correct Answer:  He shows sad feelings and expression when

he described the financial collapse of the company which adds to the sense
of empathy that video describes.

- Question (visual reasoning):  What was shown to support the effects of a
high cholesterol diet in the video?  Correct Answer:  Video demonstrates
how cholesterol gradually clogs blood vessels, using an animation to
illustrate the cross-section of vessels and the buildup of plaque.

Bad example questions because it can be answered by common sense.  -
Question (counting):  How many players are there in a soccer team?  Correct
Answer:  11.

Bad example questions because it asks for trivial details.  - Question
(counting):  How many times the word 'hurricane' is said in the video?
Correct Answer:  7.

Bad example questions because the summary of topics are subjective and
ambiguous.  - Question (timeline event):  List the sequence of topics
Grace discusses in the video, starting with the earliest.  Correct Answer:
Getting ready for a photoshoot, attending a baseball game, showing off her
new outfit, playing a Wayne's World board game, and discussing her upcoming
week.

Dense Caption with Timestamps:  {video_inputs_str}

### C.2.4  GENERATING DECOYS FROM QUESTIONS AND ANSWERS

Role:  You are a rigorous teacher in a "Long-term Video Understanding"
class.  You will assist students in developing strong critical thinking
skills.  This requires creating sophisticated test questions to accompany
video content.

Understanding:  I will provide:

- Dense Captions:  A breakdown of the video, including structure, key
events, and timestamps.  - Target Questions & Answers:  A set of
{target_number} questions about the video, along with their correct answers.

Task:  Generate High-Quality Multiple-Choice Questions

1.  Analyze:  Carefully study the dense captions, questions, and correct
answers.  Familiarize yourself with the nuanced details of the video
content.

2.  Decoy Design:  For each target question, generate {decoy_number}
incorrect answers (distractors).  These distractors must be:
- Challenging:  Plausible to the point where students need deep content
understanding and critical thinking to choose the correct answer.
- Stylistic Match:  Mimic the style, tone, and complexity of the correct
answer.
- Similar Length:  Keep length close to that of the correct answer,
preventing students from eliminating choices based on length differences.
- Factually Relevant:  Related to the video content, even if slightly
incorrect due to a detail change, misinterpretation, or logical fallacy.
- Reasonable:  Each decoy should be something that could be true, making
simple elimination impossible.

Specific Techniques for Distractor Creation

- Subtle Tweaks:  Alter a minor detail from the correct answer (e.g., change
a time, location, or name).
- Confusing Similarity:  Use a concept from elsewhere in the video that
seems related but applies to a different context.
- Misdirection:  Introduce a true statement related to the video's theme but

not directly answering the question.
- Order Shuffling:  If the question involves the order of events, subtly rearrange the order within the distractors.

Presentation:

- QUESTION: Repeat the provided question faithfully (e.g., "QUESTION 1 (Capability):  ...")
- CORRECT ANSWER: Repeat the correct answer (e.g., "CORRECT ANSWER: ...")
- WRONG ANSWERS: List each wrong answer on a separate line without using letters to label choices (e.g., "WRONG ANSWER 1:  ...", "WRONG ANSWER 2: ...")

*GOOD* Example:  Question:  What are the three main challenges that the college is taking on?  Correct Answer:  Food scarcity, pollution, and disease.  Wrong Answer 1:  Global warming, deforestation, and poverty. Wrong Answer 2:  Hunger, homelessness, and crime.  Wrong Answer 3:  Obesity, malnutrition, and food insecurity.  Wrong Answer 4:  Food waste, water shortages, and air pollution.

*BAD* examples where the decoys format is different from correct answer: Question:  What color is the shirt that the woman is wearing?  Correct Answer:  Black.  Wrong Answer 1:  The woman is wearing a white shirt. Wrong Answer 2:  The woman is wearing a blue shirt.  Wrong Answer 3:  The woman is wearing a green shirt.  Wrong Answer 4:  The woman is wearing a red shirt.

*BAD* examples because only the correct answer is in positive sentiment. Question:  What is the overall sentiment of the man in the video?  Correct Answer:  He is overjoyed with his new gift.  Wrong Answer 1:  He is upset his gift is not big enough.  Wrong Answer 2:  He is sad about life in general.  Wrong Answer 3:  He is upset the gift is not great.  Wrong Answer 4:  He seems down and unhappy.

Dense Caption with Timestamps:  {video_inputs_str}

Question and Correct Answer:  {question_and_answer_str}

### C.2.5  QAD FILTERING

The following prompt is used to filter out questions that can solve from QADs alone.

Instructions:

Carefully analyze the following question and options.  Rank the options provided below, from the most likely correct answer to the least likely correct answer.  Please respond with "ANSWER" and "EXPLANATION".

Your response should be in the following format:
* ANSWER: [Letter of the ranking, split by greater than symbol.  (e.g., "ANSWER: A > B > C > D > E")].
* EXPLANATION: [Provide a brief explanation of your choice.  Do not repeat the option.]

QUESTION: {question_str}

Options:  {options_str}

Please provide your response below.

### C.3  HUMAN RATING AND CORRECTION OF QADS

We provide a screenshot of the UI used by raters to annotate automatically generated QADs in Fig. 10. Note that if any of the four options under the 'Is the question valuable' field are not selected,

then the question is discarded from the dataset. We made sure to train raters using training raters (with detailed decks and feedback rounds), as well as applying rater replication (we used 3 raters per question independently), and rater pipelining (having an experienced rater verify the answer from a previous rater) in order to correct hallucinations and other mistakes, and discard QADs that were inappropriate. Overall, of the total 11,030 QADs that we obtained automatically, 7,762 ( 70%) were discarded by raters.

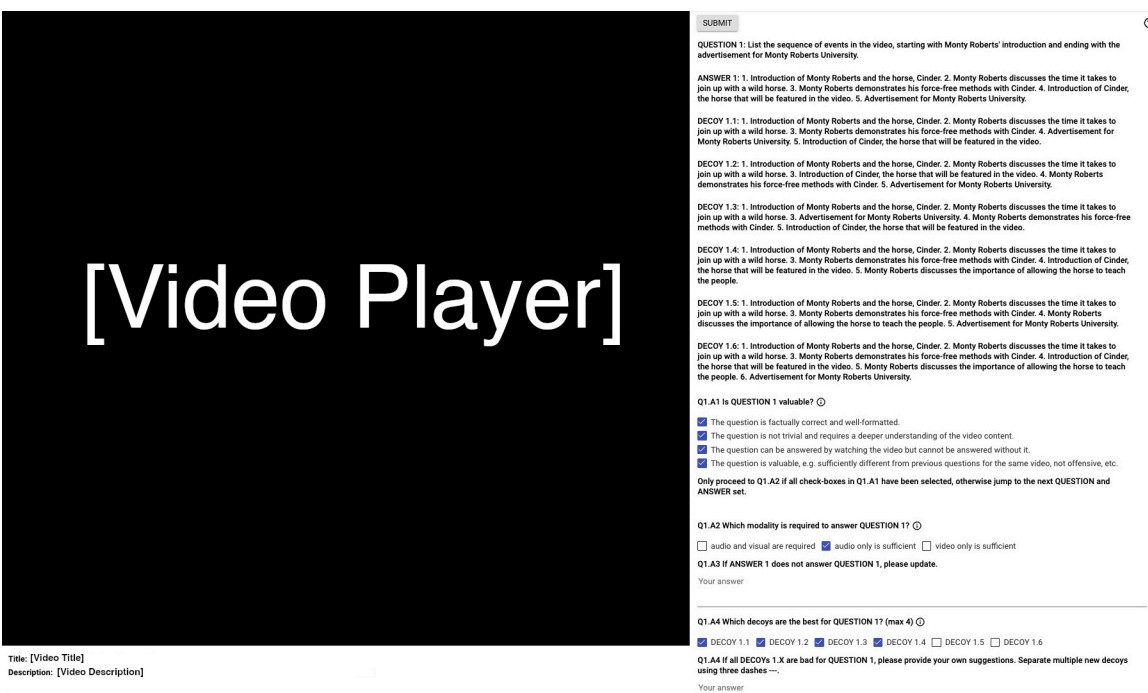

Figure 10: **Screenshot of rater UI.**

## C.4 FILTERING SUBSETS

Here we provide details for how we select the thresholds used to create the NEPTUNE-MMH and NEPTUNE-MMA subsets. For both subsets, we filtered NEPTUNE-FULL with the QAD filter described in Sec. 4.4. For NEPTUNE-MMA, we additionally filtered out QADs that human raters marked as requiring only the audio modality and answer (see Sec. 4.5). We refer to this as the "rater test". For NEPTUNE-MMH, we instead applied the ASR filter (Sec. 4.4). Both QAD and ASR filters were run by prompting an LLM (Gemini 1.0 Pro) three times, each time with a different random seed and then removing QADs that the LLM answered correctly at least X out of three times, where X is the threshold for the test.

Fig. 11 shows how choosing different thresholds affects dataset size and accuracy scores. The top row shows the choices for the NEPTUNE-MMH subset. Raters marked almost half of the questions as answerable from audio only, so the rater filter already cuts the dataset size in half. Successively applying the QAD filter with increasing thresholds reduces data size up until less than 25%. We benchmark three models on the different subsets that have access to ASR only, vision only, or both vision and ASR, respectively. As expected, all three models show declining performance, with the ASR-only model showing the biggest losses. This suggests that all models were inferring the correct answer from the QAD only, which the filter successfully mitigates. The vision-only model gains slightly from removing QADs that fail the rater rest, which is expected as the test removes QADs that rely on audio, which the model does not have access to. However, like for the other models, its accuracy declines when adding the QAD test.

The bottom row of Fig. 11 shows the choices for the NEPTUNE-MMA subset where we use the ASR filter and the QAD filter with identical thresholds. This filter set has a stronger effect on the dataset size, reducing it to less than 15% of its original size at the highest threshold. Because the ASR-only

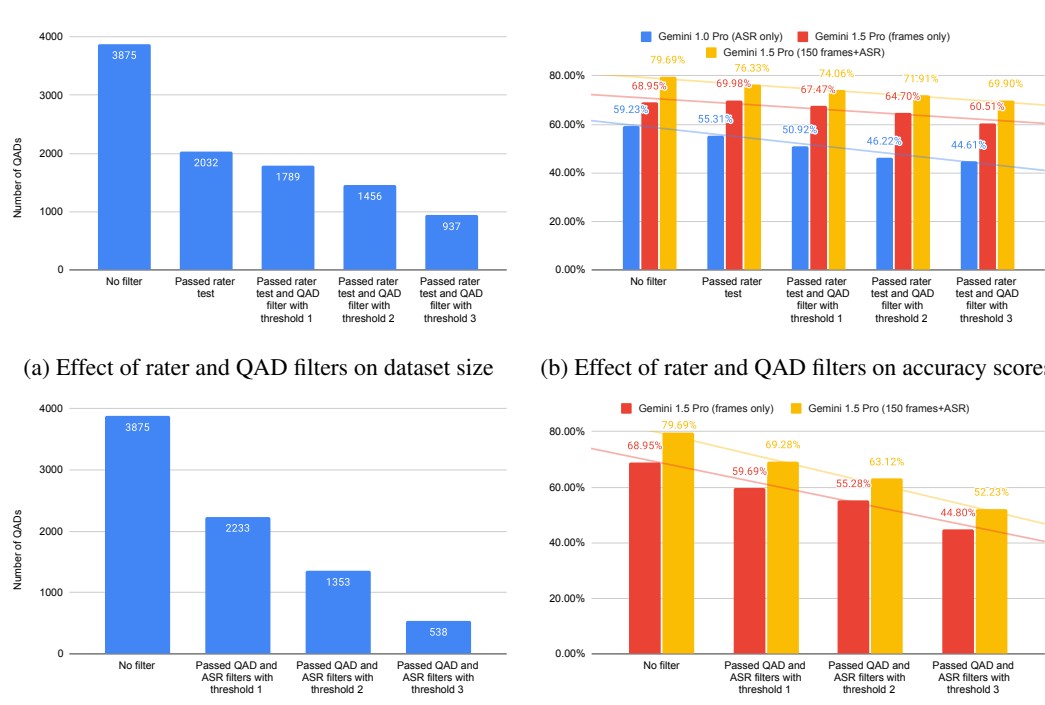

(a) Effect of rater and QAD filters on dataset size

(b) Effect of rater and QAD filters on accuracy scores

(c) Effect of ASR and QAD filters on dataset size

(d) Effect of ASR and QAD filters on accuracy scores

Figure 11: Effect of filtering thresholds for the NEPTUNE-MMH (top row) and NEPTUNE-MMA (bottom row) subsets.

model was used for the ASR filter, we exclude it from the accuracy comparison. The vision-only and vision+ASR models both show declining accuracy with increasing thresholds. As expected, the accuracy of the vision+ASR model declines faster. The effect of this filter set on the accuracy is much stronger than that of the above filter set, suggesting that it increases the difficulty of the dataset more strongly. Even the vision-only model declines faster than above, suggesting that this filter set generally removes easier questions, even those that rely on vision only.

For both filtered sets, we opted to set the threshold to two, which in both cases significantly increases the dataset difficulty while still preserving enough QADs for statistically meaningful evaluation metrics. We noticed that when setting the threshold to three, there were less than five QADs left for some question types, preventing robust accuracy estimation for these tasks.

## C.5 IMPLEMENTATION DETAILS FOR BENCHMARKS

### C.5.1 BLIND BASELINES

For the Gemini-1.5-pro baseline with text only the prompt used was: "Carefully analyze the question and all available options then pick the most probable answer for this question"

### C.5.2 VIDEO-LLAVA

For Video-LLaVA the following prompt was used - "Pick a correct option to answer the question. Question: question Options: options ASSISTANT:".

### C.5.3 VIDEOLLAMA2

During inference, we uniformly sampled 8 frames from each video. Each frame undergoes padding and resizing to a standardized dimension. The pre-processed frames are then fed into the image encoder. These steps are set as default in the inference script provided by videoLlama2.

**QAD Prompt:** _PROMPT_TEMPLATE = """Pick a correct option number to answer the question. Question: {question} Options: {options}:"""

**OE Prompt:** Question: {question}

Output post processing: We eliminated extra characters and spaces using regex to get the final ID of the predicted option.

### C.5.4 MiniGPT4-Video

We set the 300 maximum number of output tokens to be 300 for the open-ended task and 10 for the multiple choice eval. The prompts are as follows:

_PROMPT_TEMPLATE_MCQ = """Question: select the correct option for this task: question Options: options. Output format: [OPTION]: [Reason]"""

_PROMPT_TEMPLATE_OPEN_ENDED = """Question: question Answer:"""

### C.5.5 MA-LMM

We set the 300 maximum number of output tokens to be 300 for the open-ended task and 300 for the multiple choice eval. The prompts are as follows:

_PROMPT_TEMPLATE_MCQ = """Question: select the best choice for this task: question Options: options Answer:"""

_PROMPT_TEMPLATE_OPEN_ENDED = """Question: question Answer:"""

### C.5.6 GPT-4o prompts

**Open-ended evaluation with transcript**

```
You are an expert in video understanding and question answering.  You can
analyze a video given its image sequence and and transcript and answer
questions based on them.

{video_frames}

Video Transcript:  {transcript}

Answer the question using the image sequence.  Do not describe the frames
just answer the question.  Question:  {question}
```

**Open-ended evaluation without transcript**

```
You are an expert in video understanding and question answering.  You can
analyze a video given its image sequence and answer questions based on
them.

{video_frames}

Answer the question using the image sequence.  Do not describe the frames
just answer the question.  Question:  {question}
```

**Multiple-choice evaluation with transcript**

```
You are an expert in video understanding and question answering.  You can
analyze a video given its image sequence and and transcript and answer
questions based on them.

{video_frames}

Video Transcript:  {transcript}

Answer the question using the image sequence.  Do not describe the frames
just answer the question by identifying the choice.  Question:  {question}
Choices:  {choices} Please identify the correct CHOICE and explain your
reasoning concisely.  Output Format:  [CHOICE]: [REASON]
```

**Multiple-choice evaluation without transcript**

```
You are an expert in video understanding and question answering.  You can
analyze a video as an image sequence and answer questions based on that.

{video_frames}

Answer the question using the image sequence.  Do not describe the frames
just answer the question by identifying the choice.  Question:  {question}
Choices:  {choices} Please identify the correct CHOICE and explain your
reasoning concisely.  Output Format:  [CHOICE]: [REASON]
```

### C.6   COMPUTE RESOURCES

The compute heavy part of the project was image frame captioning (as this involves reading high dimensional pixel data). The rest of the pipeline involves largely text-only LLMs and hence was less compute heavy. We estimate that the entire project in total took roughly 256 TPU v5e running over a period of 50 days.

## D   ADDITIONAL DETAILS FOR GEM

### D.1   CREATION OF GEM EQUIVALENCE DEV SET

To create a development set that allows us to estimate the accuracy of different open-ended question answering metrics on Neptune, we sampled 97 question-answer pairs from the dataset and generated 3 candidate answers per question by prompting VideoLLAVA (Lin et al., 2023), Gemini-1.5-pro (Reid et al., 2024) and MA-LMM (He et al., 2024b) to write a free-form answer for each question without looking into the decoys or ground truth. We then manually annotated these responses between 0 and 1 by comparing it to the ground truth answer. We made sure that the annotators are blind to the model to avoid any bias. The resulting set has 292 equivalence pairs with an average score of 0.32, with 85 examples having score greater 0.5 and 206 examples with score less than 0.5

### D.2   BENCHMARKING ON THE DEV SET

In Table. 1, we evaluate several open-ended metrics on our dev set. The task of the metric is to classify whether the open-ended response and ground-truth answer are equivalent or not. We report F1-scores to balance false-positives and false-negatives. We evaluate both traditional rule-based metrics such as CIDEr and ROUGE-L, as well as established model-based metrics such as BEM(Bulian et al., 2022). We also try using Gemini-1.5-pro (Reid et al., 2024) as an LLM based equivalence metric (by prompting it to estimate equivalence). First, we note that as expected, Gemini-1.5-pro correlates well with the human ground-truth annotation of the set, achieving a high F1-score of 72.5. However, given that Gemini is not open-source and proprietary, any change in the model can affect all the prior results in an external leader-board making it challenging as a metric. Traditional rule-based metrics perform much worse than Gemini-1.5-pro on this dev set as they are n-gram based and struggle to handle the diversity of domains and styles in the open-ended responses. The BERT model based BEM metric (Bulian et al., 2022) performs similarly, achieving an F1-score of 61.5.

Next, we evaluate lightweight open-source language models Gemma-2B (Team et al., 2024a), Gemma-7B (Team et al., 2024a) and Gemma-9B (Team et al., 2024b) in a zero-shot setting and find that performance improves with model size, with Gemma-9B bridging the gap well between traditional metrics and the Gemini-1.5-pro based metric. Finally, we fine-tune Gemma-9B on the open-source BEM answer equivalence dataset (Bulian et al., 2022), and find that Gemma-9B finetuned on the BEM dataset performs the best on our dev-set. We name this metric *GEM*.

### D.3   IMPLEMENTATION DETAILS

We use instruction-tuned variants of the Gemma models (gemma-it-2b, gemma-it-7b and gemma-it-9b) for our experiments. To develop a prompt, we experiment with several variations in a zero-shot setting and measure the performance on the dev-set. Our final prompt is shown below. To ensure responses occur in a standard format, we simply measure the softmax-probability over "TRUE"

response indicating the statements are equivalent and "FALSE" response indicating the statements are not equivalent. For each model, the threshold over probability is chosen to maximize the F-1 score on dev set. To finetune Gemma models on BEM dataset, we tokenize the same prompt as used in the zero-shot setting and train it using prefix-LM tuning for 10000 iterations using a learning rate of $1e - 6$. For evaluation, we truncate the open-ended responses to 100 words, use a decode cache size of 1024 and threshold the softmax probability of the LM using the chosen threshold from dev-set.

```
<start_of_turn>user
Answer Equivalence Instructions:

Carefully consider the following question and answers.
You will be shown a "gold-standard" answer from a human annotator,
referred to as the "Reference Answer" and a "Candidate Answer".
Your task is to determine whether the two answers are semantically
equivalent.

In general, a candidate answer is a good answer in place of the "gold"
reference if both the following are satisfied:
  1. The candidate contains at least the same (or more) relevant information
     as the reference,taking into account the question; in particular it
     does not omit any relevant information present in the reference.
  2. The candidate contains neither misleading or excessive superfluous
     information not present in the reference, taking into account the
     question.

Your response should be one word, "TRUE" or "FALSE", in the following
format:
ANSWERS_ARE_EQUIVALENT: [TRUE or FALSE]

Question:
"{}"

Candidate Answer:
"{}"

Reference Answer:
"{}"

Please provide your response below.
<end_of_turn>
<start_of_turn>model
ANSWERS_ARE_EQUIVALENT:
```

### D.4 QUALITATIVE EXAMPLES FOR METRIC

Below, we provide some qualitative examples of our metric applied to the predictions provided by the baseline models Video LLaMA2(Cheng et al., 2024a), Gemini-1.5-flash (Reid et al., 2024) and Gemini.1.5-pro (Reid et al., 2024). We find that Gemini-1.5-flash responses are qualitatively slightly better than Gemini-1.5-pro responses as they contain less hallucinations. Furthermore, we find that with a single frame as input, Gemini-1.5-pro usually responds with a variation of "This information is not present is the provided frame" instead of making an educated guess. This is unlike the case of multiple choice questions where the model commits to one of the choices. These aspects are reflected in the results presented in Table.4 and Table.3. We also find that the metric fails often in the questions about listing items / events in order. In such cases, the model is biased towards accepting the response (false positives) even if the order is incorrect provided the items / events are present in the candidate response.

```
Question: Compare the Pampered Chef Veggie Strip Maker to a food
    processor in terms of their effectiveness in making cauliflower
```

rice.
Answer: The Veggie Strip Maker is more effective than a food
    processor in making cauliflower rice because it prevents the
    cauliflower from becoming mushy.
Video LLaMA2: Both the Pampered Chef Veggie Strip Maker and a food
    processor can be effective in making cauliflower rice, but they
    differ in their methods of operation and the resulting texture of
    the rice.   The Pampered Chef Veggie Strip Maker is a kitchen
    appliance designed specifically for making vegetable strips,
    including cauliflower rice. It uses a grating mechanism to create
    rice-like pieces by shredding the cauliflower. This method can
    produce a relatively uniform and rice-like texture.   On the other
    hand, a food processor is a versatile kitchen appliance that can
    also be used to make ...
GEM Equivalence: False
Gemini-1.5-flash: The Pampered Chef Veggie Strip Maker makes
    cauliflower rice that is less mushy than a food processor.
GEM Equivalence: True
Gemini-1.5-pro: The Pampered Chef Veggie Strip Maker is more
    effective than a food processor because it doesn't make the
    cauliflower mushy.
GEM Equivalence: True

Question: How does the use of behavioral training help in the
    process of taking foot x-rays on elephants?
Answer: Behavioral training encourages elephants to willingly
    participate in their healthcare, making the process of taking foot
    x-rays less stressful for both the elephants and the caretakers.
Video LLaMA2: The use of behavioral training plays a crucial role in
    the process of taking foot x-rays on elephants. By training the
    elephants to follow commands and stand on a platform, it allows the
    caretakers to safely and effectively capture the x-rays of the
    elephants' feet. This training helps to ensure that the elephants
    remain still and cooperative during the x-ray process, minimizing
    any potential harm or discomfort. Additionally, the behavioral
    training helps to build a positive relationship between the
    elephants and their caretakers, fostering trust and cooperation.
    Overall, the use of behavioral training enhances the accuracy and
    efficiency of the
GEM Equivalence: True
Gemini-1.5-flash: Behavioral training allows the elephants to
    willingly participate in the training sessions and participate in
    their own health care.
GEM Equivalence: True
Gemini-1.5-pro: Behavioral training allows the elephants to
    willingly participate in the x-ray process by presenting their feet
    for imaging.
GEM Equivalence: True

Question: What are the key features of the Adam Purvis Progeny
    knife's blade?
Answer: The Adam Purvis Progeny knife's blade is made of 3.4-inch
    M390 steel. The blade shape is useful, simple, and highly effective,
    the edge is very sharp, featuring a stonewash finish on the primary
    bevels and a horizontal satin on the flats. It is exceptionally
    sharp and has a 150 thousandths stock thickness, making it suitable
    for everyday carry tasks.
Video LLaMA2: The Adam Purvis Progeny knife's blade is made of S35VN

steel, which is known for its high strength and excellent edge
retention. The blade has a flat ground and a hollow grind, which
allows for a thin and sharp edge. Additionally, the blade has a
black and silver color scheme, giving it a sleek and modern
appearance.
GEM Equivalence: False
Gemini-1.5-flash: The Adam Purvis Progeny knife's blade is made of
M390 steel, has a modified sheepsfoot style, and is ground very
thinly for a sharp edge.
GEM Equivalence: False
Gemini-1.5-pro: The Adam Purvis Progeny knife features a 3.4-inch
M390 steel blade with a modified sheepsfoot shape, thin grind, sharp
edge, and a stonewashed finish on the bevels with a satin finish on
the flats.
GEM Equivalence: True

Question: How does interacting with civilians in nearby villages
enhance cadets' leadership skills?
Answer: Engaging with civilians in nearby villages allows cadets to
develop respectful communication and relationship-building skills,
which are essential for effective leadership in various situations.
Video LLaMA2: Interacting with civilians in nearby villages enhances
cadets' leadership skills by providing them with real-world
experiences and opportunities to apply their knowledge and training
in a practical setting. By engaging with the local community, cadets
learn to communicate effectively, build trust, and foster positive
relationships with people from diverse backgrounds. This hands-on
experience helps them develop their interpersonal skills, cultural
awareness, and problem-solving abilities, all of which are crucial
for successful leadership in various situations. Additionally,
interacting with civilians allows cadets to gain a deeper
understanding of the challenges and needs of the community, which
can inform their decision-making and ...
GEM Equivalence: True
Gemini-1.5-flash: Interacting with civilians in nearby villages
teaches cadets how to speak to people respectfully and build trust.
GEM Equivalence: True
Gemini-1.5-pro: It teaches them how to speak to people.
GEM Equivalence: False

Question: List the sequence of topics discussed in the video,
starting with the earliest.
Answer: 1. Introduction of the GAN356 AIR cube and its unique
features. 2. Discussion of the GAN356 AIR's weight and GES (Gan
Elasticity System) springs. 3. Lubrication, tensioning, and personal
experience with the GAN356 AIR's performance. 4. Assessment of the
GAN356 AIR's performance, including its smoothness, speed, corner-
cutting, and tendency to lock up. 5. Overall evaluation of the
GAN356 AIR, including its strengths and weaknesses, and
recommendations for potential buyers.
Video LLaMA2: 1. Introduction to the video and the Rubik's Cube. 2.
Explanation of the different types of Genuine Handcubes. 3.
Demonstration of how to solve the Rubik's Cube, including the
various algorithms and techniques used. 4. Showcasing of the
different colors and types of Rubik's Cubes available. 5.
Conclusion and closing credits.
GEM Equivalence: False

```
Gemini-1.5-flash: The video starts with an introduction of the GAN
    356 Air cube, then discusses its features, including the GES system
    and its light weight. The reviewer then shares their opinion on the
    different GES springs and their effect on the cube's feel. The video
    continues with a discussion on the cube's performance, including its
    smoothness, speed, and tendency to lock up. Finally, the reviewer
    concludes with a recommendation for viewers who might be interested
    in purchasing the cube.
GEM Equivalence: True
Gemini-1.5-pro: The video discusses the main features of the Gan 356
    Air, the different types of springs, the weight and price of the
    cube, the reviewer's experience with the cube, and the reviewer's
    overall opinion of the cube.
GEM Equivalence: False

Question: In what order are the following events mentioned in the
    video: introduction of the wallet, features of the wallet, and care
    instructions?
Answer: Introduction of the wallet, features of the wallet, care
    instructions.
Video LLaMA2: The video begins with the introduction of the wallet,
    followed by a description of its features, and concludes with
    instructions on how to care for the wallet.
GEM Equivalence: True
Gemini-1.5-flash: The speaker introduces the wallet, then describes
    its features, and finally mentions the care instructions.
GEM Equivalence: True
Gemini-1.5-pro: Introduction of the wallet, care instructions,
    features of the wallet.
GEM Equivalence: False
```

# E    MODEL-BASED TEMPORAL CERTIFICATE

The idea of a temporal certificate was introduced by EgoSchema (Mangalam et al., 2023) as a way of capturing the intrinsic temporal understanding length for *long video* QA datasets. It is defined as 'the length of the video a human verifier needs to observe to be convinced of the veracity of the marked annotation'. While the authors used it to uncover flaws in existing long video QA datasets, as well as to provide a difficulty measure independent of video length, we find that is has the following drawbacks: (i) it does not take into account the *length of time* or the *effort* taken by the annotator themselves, to find the correct time span in videos; (ii) it requires manual annotation from expert annotators to measure; and finally (iii) is subjective.

As an attempt to mitigate these issues, we introduce a slightly modified version of the temporal certificate, which is *Model-Based*. We calculate this certificate using 129 samples from Neptune and EgoSchema, respectively. For this experiment we used Gemini 1.5 Pro, with one "driver" model run to answer the question and two other model runs with different random seeds to verify if the answer was not correct by random chance. Along with the question and options, we provided video clips of various lengths from the center of the video, and at various fps, as shown in Fig. 12.

Since this experiment queried a set of frames over various clip lengths, we defined it as the "needle in haystack" problem. Here, the needle is defined as a frame or set of frames needed to answer the question correctly, matching a human's ground truth response, while the haystack is a set of frames which need to be watched to find the needle frames. Iteratively, we increase the video length and fps for the query until the model achieves the correct response.

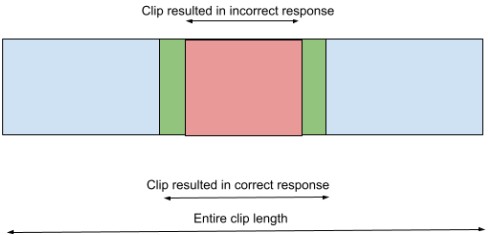

Figure 12: **Model-based Temporal Certificate:** Illustration of video clip querying for the model-based temporal certificate experiment. The red clip is the clip length that resulted in an incorrect response. As we increased the clip length wider, and the model correctly answered the question, we logged the frame count for incorrect response and correct response, and stopped querying. Besides clip length, we vary the fps of the query clip.

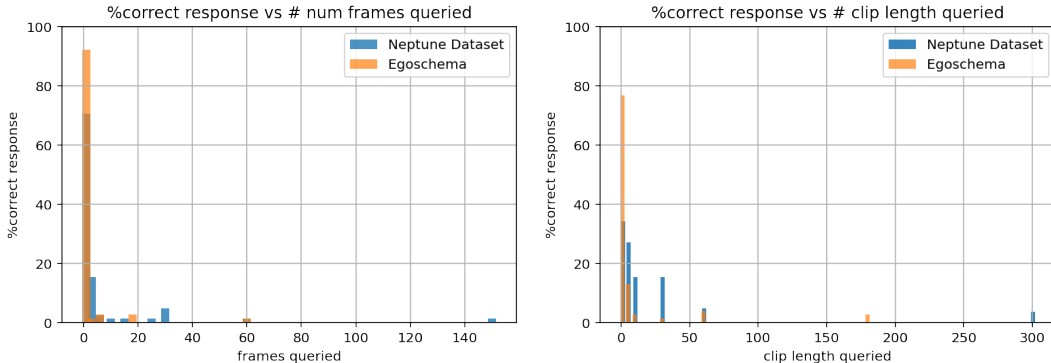

Figure 13: **Frame level temporal certificate:** We compared our dataset sample with EgoSchema to evaluate the number of frames needed by model to answer questions correctly. The figures above show the distribution of the minimum number of frames required to achieve the correct response.

As shown in Fig. 13, we find that the model needs more frames to answer the question correctly for the Neptune dataset as compared to EgoSchema. This resulted in a mean of 5.39 as certificate frames for Neptune which is 3.37 times the mean certificate frame number of 1.6 for EgoSchema. On the clip length level this translated to a mean of 21.22s of clip needed to respond correctly on the Neptune dataset, whereas for EgoSchema the mean was 9.07s. The model-based certificate lengths turn out to be much smaller than the certificate lengths reported by EgoSchema, where humans needed close to 100s to answer the questions for EgoSchema.

In addition, we define the *effort score* as the fraction of the maximum number of frames needed to be watched before answering the question correctly, as defined in Equation 1. An effort score closer to 0 suggests that the needle isn't very small compared to the haystack, i.e. most of the frames contain the answer to the question; while a high effort score means a high percentage of haystack frames needs to be included before we cover all frames required to answer correctly.

$$\text{EFFORT SCORE} = \frac{\text{MAX NUMBER OF FRAMES RESULTING IN AN INCORRECT RESPONSE}}{\text{MIN NUMBER OF FRAMES RESULTING IN A CORRECT RESPONSE}} \quad (1)$$

For Neptune, the mean effort score was 0.47, whereas for EgoSchema, it was 0.19. This suggests that Neptune requires 2.47 times the effort compared to EgoSchema according to the definition above, which closely corroborates the above results for the mean clip lengths needed to solve the questions from the respective datasets.

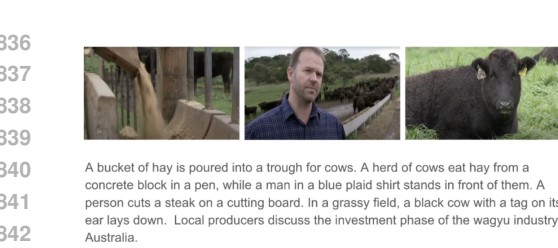

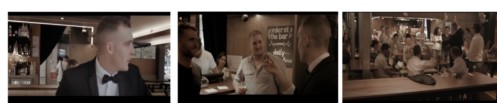

A bucket of hay is poured into a trough for cows. A herd of cows eat hay from a concrete block in a pen, while a man in a blue plaid shirt stands in front of them. A person cuts a steak on a cutting board. In a grassy field, a black cow with a tag on its ear lays down. Local producers discuss the investment phase of the wagyu industry in Australia.

Illusionist Josh Norbido attends the grand opening night of Hop & Pickle, a new venue. He waits for guests to arrive, chats with others, and explores the bar. He is seen sitting at the bar with a vodka bottle behind him, looking at signs like "order at the bar" and a McDonald's sign. The modern and stylish decor includes a mix of industrial and rustic elements, creating a warm and inviting atmosphere. The video concludes with him standing in front of an "order at the bar" sign.

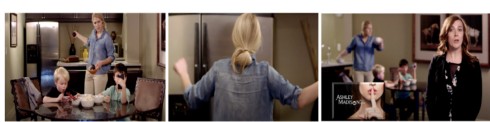

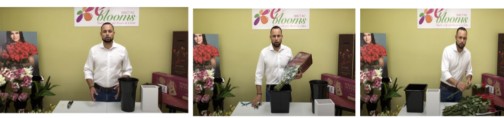

A woman expresses frustration with her family for not eating the fresh produce she buys, which often goes bad before they have a chance to consume it. She laments that she buys all this fresh produce, but nobody eats it, and wonders who she is buying it for. She suggests that her husband can use Ashley Madison to find her a new mom who appreciates healthy eating, implying that her current family does not value or consume the fresh produce she buys.

Alex Alfonzo, vice president of Blooms Direct, introduces the video in front of the "Direct Inc. Blooms" sign. He explains that the video will provide instructions on how to care for and arrange roses after receiving them

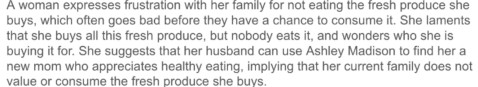

Figure 14: **Qualitative results of automatic caption generation.** Best viewed zoomed in. Note how the captions contain plenty of visual details, can contain numerous different events (top left), can mention mood and atmosphere (top right), use details from the ASR, and can even mention high level feelings and emotions (bottom left). Bottom right shows a failure case, where the caption is accurate, but too simple and high level and does not cover the fine-grained actions that the man takes.

## F  EXAMPLES OF CAPTION QUALITY

We show examples of captions generated by our automatic pipeline in Fig. 14.

## G  SOCIETAL IMPACT

Our data may match the distribution of videos and text on the internet. As such, it will mirror known biases on that source of data. For at least this reason, this data set should not be used for training models and is only intended for academic evaluation purposes. To create the dataset, we run large Gemini models, which has a negative externality of energy usage and carbon emissions. For benchmarking, we use existing models. These models are likely to inherit the biases of the data distribution and the pre-trained weights used in their original training.