# OpenReview forum: "Neptune: The Long Orbit to Benchmarking Long Video Understanding"
_ICLR.cc/2025/Conference — Submitted to ICLR 2025_

### Official Review · Reviewer_j9hH · 2024-11-02

**Soundness:** 3
**Presentation:** 3
**Contribution:** 3
**Rating:** 5
**Confidence:** 4

**Summary:**

The paper introduces NEPTUNE, a new benchmark for evaluating multimodal understanding of long videos. It addresses the limitations of current video datasets and models, which often focus on short clips and can be solved by image models applied per frame. The authors propose a semi-automatic pipeline that leverages large Video Language Models (VLMs) and Large Language Models (LLMs) to automatically generate dense, time-aligned video captions and challenging question-answer-decoy sets for video segments up to 15 minutes long. The paper also discusses the related works in video question answering, the motivation behind the NEPTUNE dataset, and the detailed pipeline for dataset creation. It concludes with the limitations of the dataset and potential areas for future improvement, such as reducing biases introduced by the models used in the creation process and enhancing the dataset with additional annotations like temporal grounding or entity labels.

**Strengths:**

1. The paper introduces an innovative semi-automatic pipeline designed to generate question-answer-decoy (QAD) sets. This method effectively reduces the annotation costs associated with manual video captioning and question generation. By leveraging large models such as VLMs and LLMs, the pipeline automates the creation of dense, time-aligned video captions, which are then used to derive challenging QAD sets for segments of video content. This approach not only scales well but also maintains a high standard of quality in the dataset.

2. A significant contribution of this paper is the training and proposal of a model for the automatic evaluation of open-ended QA responses. The authors address the limitations of existing metrics, which are either rule-based or rely on proprietary models, by finetuning an open-source model on a generic answer equivalence dataset. This results in the GEM (Gemma Equivalence Metric), a new metric that provides a more accurate assessment of open-ended responses on the NEPTUNE dataset, thus facilitating more reliable benchmarking and comparison of different models.

3. The NEPTUNE benchmark covers a wide range of video lengths and types, ensuring diversity in the dataset. This comprehensive approach guarantees that the models evaluated on this benchmark are tested on their ability to reason over a variety of video content, going beyond the capabilities required for understanding short video clips. The inclusion of different video domains and the emphasis on longer-form content make the NEPTUNE dataset a robust platform for assessing multimodal video understanding systems across a broad spectrum of real-world scenarios.

**Weaknesses:**

1. The paper does not include an evaluation and comparison with the latest open source models such as InternVL, LLaVA-OneVision, and MiniCPM. These models are part of the current research landscape and offer a different perspective on video understanding capabilities

2. The paper primarily focuses on the analysis of benchmarks like NextQA and EgoSchema but does not provide a thorough comparison with more recent benchmarks such as MLVU, Video-MME, and LongVideoBench, which are designed to evaluate long-form video understanding

3. Although the semi-automatic pipeline proposed in the paper effectively reduces the workload of manual annotation, it lacks novelty in terms of a detailed analysis and comparison with other pipelines. The paper could benefit from a more in-depth exploration of the unique aspects of its pipeline and how it compares to existing methods

4. While the proposed GEM metric may have lower evaluation costs compared to assessments using models like GPT or Gemini, the paper lacks a comparison with the consistency of human evaluations. Introducing human annotations to quantify and analyze the quality of assessments would strengthen the paper's findings

**Questions:**

Why not compare with other benchmarks and test the latest open-source models?
How to ensure the reliability of GEM's evaluation results?

---

> ### Author Response · Authors · 2024-11-23
>
> We thank the reviewer for highlighting the scalability of our pipeline, the advantages of GEM over existing methods and diversity of our benchmark\! In the following, we hope to address the reviewer’s concerns and answer their questions in the hope that the reviewer will consider raising their score.
>
> # W1: Evaluate more Open-source Models
>
> Thank you for this suggestion. As requested, we evaluate InternVL \[1\], LLaVA-OneVision \[2\] and MiniCPM-v \[3\] on Neptune and provide results below. We will add these results to the main paper as well. We also provide frame ablations for two of these models (LLaVA-OneVision and MiniCPM-v) in the Global Author Response. Note that these experiments only use the video frames, not ASR. Comparing these new results with the open source results in Tab. 4 shows that more recent open source long-context MLLMs have made significant progress and are catching up to closed source models.  We attribute these enhancements to larger training datasets and the use of the powerful Qwen-2 LLM. For reference, Gemini-1.5-Pro achieves 69.31% accuracy with 150 frames.
>
> In addition, we are planning to host a leaderboard on our Github page to keep track of the performance of newer open-source models as they are developed, and to allow authors to submit their own scores.
>
> |  | MCQ \- Neptune-Full  | MCQ \- Neptune-MMH  |
> | :---- | ----- | ----- |
> | InternVL2-8B \- 16 frames | 57.12% | 54.30% |
> | LLaVA-OneVision-7B \- 100 frames | 66.22% | 62.82% |
> | MiniCPM-v \- 50 frames  | 56.59% | 53.27% |
>
>
> \[1\]\* How Far Are We to GPT-4V? Closing the Gap to Commercial Multimodal Models with Open-Source Suites
> \[2\]\* LLaVA-OneVision: Easy Visual Task Transfer
> \[3\]\* MiniCPM-V: A GPT-4V Level MLLM on Your Phone
>
> \* (unpublished or concurrent work)
>
> # W2 & 3: Comparison to more recent benchmarks and pipelines
>
> Thank you for this suggestion\! Please see the response in the Global Author Response. We compare our pipeline in detail to 11 other prior and concurrent dataset creation pipelines, and will add this discussion to the paper.
>
> # W4: Comparing GEM to human ratings
>
> We note that the GEM dev is human-annotated, so Tab. 1 compares how well ratings of different metrics align with humans. We will make this clearer in the text. To give more details, we first manually annotate the answer equivalence dataset. The scores on this dataset are entirely provided by humans, and this human benchmarking allows us to compare the performance of GEM to human performance. On this set compared to human performance, we find our GEM metric very closely approximates the performance of the much more expensive, closed-source Gemini model. We describe this in the Global Author Rebuttal.

---

> > ### Comment · Reviewer_j9hH · 2024-11-29
> >
> > The author has solved my concern. I increase my rating to 5.

---

### Official Review · Reviewer_mqoh · 2024-11-02

**Soundness:** 3
**Presentation:** 4
**Contribution:** 3
**Rating:** 6
**Confidence:** 4

**Summary:**

This paper introduces a scalable, semi-automatic pipeline for constructing the Neptune benchmark dataset, aimed at evaluating long video understanding capabilities. A new evaluation metric, GEM, is proposed, demonstrating its advantages over traditional rule-based evaluation metrics.

**Strengths:**

1. Neptune addresses several essential question types related to temporal-aware video understanding, including the challenging temporal ordering and counting.
2.	Two subsets are introduced to comprehensively assess current multi-modal large language models.

**Weaknesses:**

1. In Figure 4, the authors show that EgoSchema reaches saturation at approximately 16 frames, while performance continues to increase with Neptune. This conclusion is drawn based on the powerful Gemini model; it would be beneficial to additionally include results from some open-source models (e.g., short or long context MLLMs) to better promote the development of open-source MLLMs.
2. Model names should be consistently formatted (e.g., VideoLLaMA2 vs. VideoLlaMA2).

**Questions:**

1. In Tables 3 and 4, it is evident that a language-only Gemini model (without ASR) even outperforms the baseline image model (BLIP2) and a short-context MLLM (Video-LLaVA). Could you provide possible reasons for this? Is it because the language models in BLIP2 and Video-LLaVA are not sufficiently robust to describe visual information?
2. In Table 3, the authors present the performance with and without ASR. The blind model (0 frame) achieves a comparable (Gemini) or even higher (VideoLLaMA2) GEM score than the model evaluated with full-frame input. Are there possible ways to mitigate the impact of ASR during dataset construction to encourage the design of vision-centric MLLMs?

---

> ### Author Response · Authors · 2024-11-23
>
> We thank the reviewer for their positive rating and constructive feedback on our paper\! In the following, we hope to address the reviewer’s concerns and answer their questions.
>
> # W1: Frame Experiments with an Open-source Model
>
> This is a great suggestion. We have run this experiment with 3 open-sourced models (VideoLLaMA2 results are in Table 3 of the main paper), and we have evaluated 2 more recent models. Results are provided in the Global Author Rebuttal. We find that Gemini saturates at around 50 frames while open-sourced models have varying saturation points, VideoLLaMA2 (8 frames), MiniCPM (50 frames) and LLaVA-OneVision (100 frames).
>
> # W2: Formatting
>
> Thank you for catching this\! We will update the paper to ensure model names are consistent.
>
> # Q1: Strong LLM
> Yes exactly, Gemini is a much bigger model and is sometimes able to use subtle cues to infer the correct answer from just the answer choices alone, which is not the case for the Flan-T5 model used in BLIP-2. This highlights the importance of a strong LLM for video understanding. We believe that the poor results with the older models are due to the small size of their language models, as suggested by the reviewer.
>
> # Q2: ASR Performance
> This is a great point, and we note that it is tricky to create datasets for long video understanding where ASR is not relevant \- other concurrent datasets such as VideoMME also show large improvements to performance when ASR is added. This is the reason we propose the hard subset (MMH), in order to highlight and assess visual understanding. MMH includes only QADs that human labelers have annotated as requiring vision to solve. On MMH there’s a clear gap between Gemini-ASR and Gemini-ASR-all-frames.

---

> > ### Comment · Reviewer_mqoh · 2024-11-26
> > **Response to authors**
> >
> > Thanks for the additional experiments and clarifications, I have no further questions. Overall, I think this benchmark is valuable for the community.

---

### Official Review · Reviewer_jd8F · 2024-11-03

**Soundness:** 3
**Presentation:** 3
**Contribution:** 2
**Rating:** 5
**Confidence:** 4

**Summary:**

The paper introduces a new benchmark dataset called Neptune, designed to assess multimodal, high-level understanding of long videos. The key contributions of the paper are as follows:

1. Semi-Automatic Pipeline: The authors propose a scalable, semi-automatic pipeline for generating challenging question-answer-decoy sets for long videos, leveraging large video language models (VLMs) and large language models (LLMs) to automatically generate dense, time-aligned video captions and tough QAD sets for video segments up to 15 minutes long.
2. Neptune Dataset: The dataset, Neptune, covers a broad range of long video reasoning abilities and includes a subset that emphasizes multimodal reasoning. It consists of 3,268 QAD annotations for 2,405 videos.
3. Evaluation Metrics: Given the limitations of existing metrics for open-ended question answering, the authors introduce a new model-based metric called GEM (Gemma Equivalence Metric) to score open-ended responses on Neptune, offering a static and reproducible evaluation method.
4. Benchmark Evaluations: The paper presents benchmark evaluations that reveal current open-source long video models perform poorly on Neptune, especially on questions of testing temporal ordering, counting, and state changes, indicating a significant gap between open-source and proprietary models like Gemini-1.5 and GPT-4.

In summary, the paper contributes a new benchmark dataset and metric aimed at advancing the field of long video understanding through multimodal reasoning, with the goal of spurring the development of more advanced models in this area.

**Strengths:**

1. The semi-automated annotation method proposed in this paper overcomes some of thechallenges associated with manual annotation, such as the creation of complex temporal reasoning questions, which can be laborious for humans. The utilization of GPT to generate these questions reduces this burden.

2. The introduction of the GEM metric addresses the need for a static, open-source evaluation metric for open-ended VideoQA, which has been a limitation in prior work.

3. The paper clearly articulates the motivation behind the Neptune dataset and the GEM metric, making it easy for readers to understand the importance of the work.

**Weaknesses:**

1. The key contributions that distinguish it from previous benchmarks are not clearly highlighted. Although the paper has made a simple comparison with EgoSchema on the impact of the number of input frames on performance, I feel that it is too simplistic. It would be constructive to see more comparisons with other existing datasets (e.g.,VideoMME) to understand how NEPTUNE's complexity and diversity align or differ from them, which could highlight the unique challenges it presents.

2. In addition to the Gemini bias discussed in the paper, I noticed that the frame captions are generated by PaLI-3. I am concerned that the hallucinations and preferences of the VLM itself (such as a preference for describing static images) may affect the quality of the
final generated QA.

3. As a benchmark for long videos, the discussion on the impact of the number of input frames on model performance is too simplistic. For example, Table 3 only discusses the performance of Gemini with 1 frame, 150 frames, and All frames. In my view, the performance with 150 frames has already reached saturation, but it is uncertain how the performance with 100 frames and 50 frames would be, that is, where the performance saturation point is.

**Questions:**

1. As a benchmark for long videos, single-frame assessment fails to reflect the significance of temporal modeling. I believe that the sensitivity to temporal order should be evaluated by adopting methods such as reversing or shuffling the input frames.

2. I have observed that the Gemma Equivalence Metric shows similar or even inferior performance compared to direct assessment with Gemini 1.5 Pro, with an accuracy rate of only around 70%. In contrast, previous works such as Lingo-Judge[1] seem to demonstrate that fine-tuned model-based evaluation methods can achieve assessment accuracy close to human judgment. Is this due to differences in dataset complexity?

3. I have noticed that the distribution of generated QA types exhibits a clear long-tail distribution, with the largest proportion being Temporal Ordering tasks. Does this not suggest that the complexity and richness of the generated questions are insufficient? In my view, Ordering tasks belong to very simple temporal reasoning and do not meet the authors' claim of being complex questions that are difficult for humans to propose.

[1] LingoQA: Visual Question Answering for Autonomous Driving

---

> ### Author Response · Authors · 2024-11-23
>
> We thank the reviewer for their thorough review and the positive comments and valuable feedback\! We try to address all points here and hope that the reviewer would consider raising their score.
>
> # W1: Contributions beyond previous benchmarks
>
> Thank you for this suggestion. Appendix A has a survey of related datasets, including a table with MLVU and other recent datasets, which we have extended to include Video-MME and LongVideoBench. We will move these to the main paper. In addition, we believe that the most significant differences are our semi-automatic pipeline (see Global Response), and the open-set evaluation with the GEM metric. Unlike Neptune, Video-MME is entirely manually annotated, and is hence harder to scale (more details in the Global Response). We also ran ablations with varying numbers of frames on Neptune and three other benchmarks. We find that most benchmarks saturate at about 50 frames, including Video-MME, which has much longer videos than Neptune. We included Perception Test here as it is new, it does not claim to be a long video benchmark and saturates at 16 frames. We added Sec. B.3 and Fig. 6 to the supplementary material to showcase these results.
>
> | \# of Frames | 1 | 16 | 50 | 100 |
> | :---- | ----: | ----: | ----: | ----: |
> | **Neptune-MMH** | 62.89 | 71.22 | **73.37** | 73.02 |
> | **Video-MME** | 56.28 | 68.22 | **70.52** | 70.05 |
> | **CinePile (2-3 mins)** | 52.32 | 56.84 | 58.97 | **59.03** |
> | **Perception Test** | 51.48 | **64.21** | 63.72 | 63.21 |
>
> # W2: Bias from Gemini and PaLI-3
>
> We agree that the efficiency improvements from using PaLI-3 and Gemini for QAD generation come with the risk of introducing model biases and hallucinations, which is why we took measures to mitigate and measure them.
>
> The main **mitigation** mechanism is running two rounds of human filtering and having every QAD reviewed by at least *four* human labelers. 65% of auto-generated questions got rejected by raters. Also, our QAD generation prompt used both the frame captions and ASR as context, so QADs are grounded in both video and audio modalities.
>
> To **measure** the impact of bias and hallucinations, we compared question-answer pairs generated by humans with those generated with our pipeline, (see “Results on HPQ and Gemini bias” starting in l. 498\) and found only a minor drop in model accuracy between the sets. While we mention only Gemini bias in this section, we should amend it to all model biases, including visual biases introduced by PaLI-3. This is because the HPQ set was annotated without any model at all, as a control for our semi-automatic data generation. We also point out that in Tab. 4, GPT-4o performs competitively with Gemini. If the data had significant biases towards Gemini, we would expect Gemini to score much higher than other models.
>
> # W3: Impact of the number of input frames
>
> We agree that it would be helpful to have a finer-grained analysis of the effect of the number of frames. We have run additional experiments towards this (see Global Response) and find the saturation point for Gemini-1.5-pro-1.0 is around 50 frames.
>
> # Q1: Sensitivity to temporal ordering
>
> Thank you for this interesting suggestion\! We will run experiments on this as well.
>
> # Q2: GEM performance
>
> This is a great point. We note that all the models applied to the dev set in Table 1 were applied zero-shot. As described in the Global Response, our dev set was created by sampling QAs from Neptune and running models to create predicted answers. The finetuning dataset (BEM) used to create GEM is not from the same domain as Neptune. In contrast, Lingo-judge \[1\] is trained on over 1000 examples and then evaluated on the same dataset, which is why we suspect the F1 score is so high. By evaluating our metric zero-shot, we hope that it can be applied to other datasets as well.
>
> \[1\] LingoQA, Marcu et al.
>
> # Q3: Question type distribution
>
> We agree that a uniform distribution of question types would be a good area for future improvement. We explain the reasons for Neptune’s current question type distribution in Sec. B.1.1 of the appendix. Summarised, the key reasons are (1) the LLM is prompted to generate these automatically, and the selection depends strongly on the given video (some question types are not possible for a video), and (2) the quality of questions produced by the LLM varies strongly by question type. Therefore, after quality checking by raters, the distribution changes significantly (we add an additional figure to the appendix showing the distribution change).
> We also found that tough temporal ordering questions are difficult to propose, as they require an identification of high level events which are non-trivial to detect. We report performance per question type, in order to judge performance across all abilities.
>
> Given the automatic nature of our pipeline, it is possible to augment the dataset with more examples of each type, which we will focus on for future work.

---

> > ### Comment · Reviewer_jd8F · 2024-11-26
> > **Response to author rebuttal**
> >
> > Thank you for your response. However, I still have the following concerns, which lead me to maintain my current score:
> >
> > 1. What are the advantages of a semi-automated process in terms of scalability? From my perspective, for a benchmark, scalability is less important than accuracy itself. Thus, I am more inclined toward manual annotation.
> >
> > 2. Regarding the imbalance in question type distribution, I still believe this is a significant issue. For instance, a model that excels at a specific type of question might achieve higher overall scores on this benchmark. The purpose of building a QA benchmark is to reflect the true capabilities of models through QA tasks. I believe that ensuring a diverse range of question types is essential to accurately capture a model's performance in real-world scenarios.

---

> > > ### Author Response · Authors · 2024-11-27
> > >
> > > We thank the reviewer for considering our response and hope to address their remaining concerns here:
> > >
> > > **1\. Scalability of semi-automatic pipeline**
> > > The semi-automated process significantly reduces the workload of human labelers. In our HPQ (human-proposed question) experiment (l. 498), we compared the rater effort of filtering and correcting automatically generated QA to the effort of writing original QA from scratch and found that our pipeline increases labeler productivity by almost 2x (see the first bullet in the Global Response for details). Note that this does not yet include writing decoy answers, which is another labor intensive step for humans. Experiments with different models on both sets (Tab. 2\) showed only minor differences in performance, suggesting that question difficulty is roughly comparable between the two approaches. Besides saving effort in generating evaluation data, the pipeline can also be used to generate training data in a more scalable way. Since data quality is less critical for training data, human effort could potentially be reduced further, or data could even be generated fully automatically, allowing for very large scale data generation.
> > >
> > > **2\. Question type imbalance**
> > > We agree that it is desirable for a benchmark to provide a holistic view of different model capabilities. Our paper includes a breakdown of model performance for different task types in Fig. 4 (bottom left). In addition, we have added a detailed table with a per-task breakdown of scores in the **new Sec. B.4 and Table 6 in the appendix**. The table shows that there are significant variations in per-task scores even for models that have roughly similar overall scores. This shows that models have strengths and weaknesses in different areas and allows an informed choice of models for different applications.
> > >
> > > We would like to note that uneven question type distribution is common among long video benchmarks. MLVU \[1\], Video-MME \[2\], MoVQA \[3\], LVBench \[4\], MMBench-Video \[5\], VideoVista \[6\], and CinePile \[7\] all have similarly skewed distributions of task types. Most benchmarks, like Neptune, report per-task scores in addition to scores averaged over all examples. MLVU \[1\] mitigates the imbalance issue in the averaged score by reporting accuracy averaged over all tasks, which gives each task type equal weight in the final score. We offer to follow this example and report the mean of per-task scores as an additional metric. Please find the resulting scores below, which we will add to the main paper.
> > >
> > > Generally, we find that task-averaged scores are slightly higher because harder question types like “temporal ordering” have higher representation in Neptune. However, we observe that the relative performance of different models is similar to the original metric.
> > >
> > >
> > > |  |  | Full |  | MMH |  |
> > > | :---- | :---- | ----- | ----- | ----- | ----- |
> > > |  | **Modalities** | **Acc** | **Task-averaged Acc.** | **Acc** | **Task-averaged Acc.** |
> > > | *Image Models* |  |  |  |  |  |
> > > | BLIP-2 | RGB (center frame) | 34.80 | 38.34 | 28.10 | 28.37 |
> > > | *Short Context MLLMs* |  |  |  |  |  |
> > > | Video-LLaVA | RGB (8 frames) | 25.79 | 30.74 | 24.00 | 24.00 |
> > > | Video-LLaMA2 | RGB (16 frames) | 44.74 | 49.51 | 40.29 | 42.61 |
> > > | Video-LLaMA2 | RGB (16 frames)+ASR | 49.28 | 56.41 | 45.38 | 53.22 |
> > > | *Long Context MLLMs* |  |  |  |  |  |
> > > | MA-LMM | RGB (120 frames) | 20.22 | 19.49 | 19.51 | 20.79 |
> > > | MiniGPT4-Video | RGB (45 frames) | 24.63 | 27.43 | 22.89 | 28.88 |
> > > | LLaVA-OneVision | RGB (100 frames) | 66.22 | 70.20 | 62.82 | 66.73 |
> > > | MiniCPM-V 2.6 | RGB (50 frames) | 56.59 | 62.53 | 53.27 | 57.30 |
> > > | *Closed-source MLLMs* |  |  |  |  |  |
> > > | VLM captions \+ LLM (JCEF) | VLM captions (16 frames) | 58.51 | 60.26 | 56.45 | 53.56 |
> > > | GPT-4o | RGB (8 frames)+ASR | 80.23 | 83.31 | 72.86 | 77.89 |
> > > | Gemini-1.5-pro | RGB(all frames) \+ ASR | 80.66 | 84.29 | 75.32 | 80.45 |
> > > | Gemini-1.5-flash | RGB(all frames) \+ ASR | 76.90 | 80.37 | 71.05 | 75.58 |
> > >
> > > \[1\] MLVU, Zhou et al.
> > > \[2\] Video-MME, Fu et al.
> > > \[3\] MoVQA, Zhang et al.
> > > \[4\] LVBench, Wang et al.
> > > \[5\] MMBench-Video, Fang et al.
> > > \[6\] VideoVista, Li et al.
> > > \[7\] CinePile, Rawal et al.

---

> > > > ### Author Response · Authors · 2024-12-02
> > > >
> > > > Dear reviewer!
> > > >
> > > > Please let us know if our additional responses have addressed your remaining concerns. We'd be happy to answer any other questions you might have.
> > > >
> > > > Best,
> > > > Neptune authors

---

> ### Comment · Reviewer_jd8F · 2024-12-03
>
> Thank you for your response, which has alleviated some of my concerns regarding the imbalance in question types. However, I believe that the inherent issue of question-type imbalance within the benchmark still persists. While adjusting the metrics can mitigate the risk of individual tasks disproportionately influencing the overall score, it may also compromise the robustness of testing for other tasks. Specifically, an insufficient number of questions could affect the evaluation accuracy for certain task types. Additionally, I maintain my reservations about the semi-automatic benchmark construction process. If this pipeline could be extended to produce high-quality long-video instruction-tuning datasets (e.g., Vript[1], CinePine[2]), it would better demonstrate its effectiveness and contribute more significantly to the community. In summary, I stand by my previous rating.
>
> [1] Vript: A Video Is Worth Thousands of Words
>
> [2] CinePile: A Long Video Question Answering Dataset and Benchmark

---

> > ### Author Response · Authors · 2024-12-04
> >
> > Thank you for your response\! We are glad that we could mitigate some of your concerns about the data distribution. We'd like to provide some final comments:
> >
> > 1\. **Pipeline scalability**: We would like to clarify that our pipeline can be used to generate instruction tuning data without modification, but leave this for future work.
> > 2\. **Data distribution**: We acknowledge that score computation is less robust for tasks with fewer examples. We leave generating more a balanced set of questions as future work, but offer to summarize the task types with lower question counts such that no task has fewer than 100 questions, which will give robust estimates of model performance for all tasks. Concretely, we summarize "Creator intent", "Predictive", "Goal reasoning" under the "Predictive" type which then has 114 questions. We also summarize "Identification" and "Comparison" into "Identification and Comparison" with 259 questions.
> >
> > We hope that this would alleviate the reviewer's final concerns.

---

### Author Response · Authors · 2024-11-23

We thank the reviewers for their insightful comments, highlighting the strengths of our dataset and suggesting areas for improvement. Here, we address points raised by multiple reviewers.

# Comparison to other Long Video dataset pipelines

We highlight novelties of our dataset, and its creation pipeline compared to both previous and concurrent works \[1-11\].

1. **Reducing Manual Effort**: Our pipeline is semi-automatic, unlike manual pipelines \[4,5,3,11\]. In LVBench \[3\] and VideoMME \[11\], even the video selection is done manually, and for MoVQA \[1\], only the decoys are generated automatically. We asked raters to propose QAs (but not decoys) for videos from scratch, and found the time taken (19.03 minutes), longer than simply discarding or correcting QAs generated automatically (10.32 minutes), i.e. showing our pipeline almost halves rater effort. Manually proposing decoys is even more challenging, both in time as well as effort (as it requires more creativity to come up with incorrect answers). The closest to our pipeline is concurrent work VideoVista \[6\], which uses GPT4 to generate QADs automatically. However the performance of GPT4 and Gemini-1.5 on this dataset is close to saturated (98% on some categories).
2. **Scalability to generic YouTube videos**: Our pipeline can be applied to any generic YouTube video. This is unlike EgoSchema \[2\], which relies on human generated captions, SFD \[9\], which requires movie titles, loglines and synopses (human-written), or MLVU \[4\], which re-uses annotations from existing datasets for many of their tasks. This makes the dataset scalable, as YouTube has a constantly growing set of videos. Potential further use cases could be applying this pipeline to generate large quantities of training data.
3. **High Quality Video Captions using careful multi-stage prompting**. A key novelty is the ability to generate high quality video captions, using a multi-stage process (Sec 4.2) which involves generating shot-level captions, clustering shots into segments, and adding visual support captions. Examples of caption quality are provided in the supplementary (Fig. 13 and Sec. F), showcasing details such as visual support, numerous events, mood and atmosphere, details from the ASR, and even high level feelings and emotions.

# Clarifications on the GEM metric

***Training:*** We train GEM by fine-tuning the open source Gemma model on BEM, a dataset of general answer equivalence ratings. Therefore, GEM is not specific to Neptune and generally applicable to other answer equivalence problems as well.
***Comparison to human assessments:*** To assess GEM (Sec. 5.1, appendix D.1, D.2), we **manually annotate** an answer equivalence dataset by sampling 97 question-answer pairs from Neptune. We then generate 3 candidate answers per question by prompting VideoLLAVA, Gemini-1.5-Pro and MA-LMM to write a free-form answer for each question without looking into the decoys or ground truth. We then used human raters to manually score these responses, resulting in a dev set with 291 QA-pairs that allows us to compare GEM to human performance. GEM metric reaches an F1-Score of 71.2, which very closely approximates the performance of the much more expensive, closed-source Gemini model (F1-Score 72.8). We will move some details from the appendix to the main paper.

# Additional Frame Experiments

We provide additional frame experiments for both Gemini-1.5-Pro and new open-sourced models below. We find that Gemini saturates at around 50 frames while open-sourced models have varying saturation points, VideoLLaMA2 (8 frames), MiniCPM (50 frames) and LLaVA-OneVision (100 frames). For both MiniCPM and LLaVA-OneVision we were unable to fit more frames into the context window.

This is an interesting experiment which we will add to the paper. Note that these are uniformly sampled frames, and hence cover the full span of the video.

| \# of Frames | 1 | 8 | 16 | 50 | 100 | 150 |
| :---- | ----: | ----: | ----: | ----: | ----: | ----: |
| **Gemini-1.5-Pro** | 55.57 | 63.80 | 66.62 | 70.08 | **70.44** | 69.31 |
| **LLaVA-OneVision**  | 56.56 | 62.51 | 63.98 | 65.70 | **66.22** | Out of context |
| **MiniCPM-V 2.6**  | 50.89 | 53.83 | 53.86 | **56.59** | 55.00 | Out of context |
| **VideoLLaMA2** | 40.88 | **44.74** | 44.74 | \- | \- | \- |

We also provide frame ablations comparing Neptune to other datasets in our response to reviewer jd8F.

\[1\]\* MoVQA, Zhang et al.
\[2\] EgoSchema, Mangalam et al.
\[3\]\* LVBench, Wang et al.
\[4\]\* MLVU, Zhou et al.
\[5\] MMBench-Video, Fang et al.
\[6\]\* VideoVista, Li et al.
\[7\]\* ReXTime, Chen et al.
\[8\]\* LongVideoBench, Wu et al.
\[9\]\* Short Film Dataset, Ghermi et al.
\[10\] Towards long-form video understanding, Wu et al.
\[11\]\* Video-MME, Fu et al.
\[12\] CinePile, Rawal et al.
\[13\] Perception Test, Pătrăucean et al.

\* (unpublished or concurrent work)

---

### Author Response · Authors · 2024-12-04
**Rebuttal Summary**

We thank the reviewers for their helpful comments and insightful questions which have helped strengthen our paper! We summarize the improvements we have made as part of the rebuttal here:

1. **Comparisons to additional open source models**: We evaluated three more recent open source models: LLaVA-OneVision, MiniCPM and InternVL2, giving a more comprehensive and up-to-date view of the state of the art on our dataset.
2. **More detailed ablation with different number of frames**: We added an ablation going from 1 to 150 frames in 6 steps. Besides Gemini, which we used in the original ablation in the paper, we also ablated on 3 open source models, providing insights into each model’s ability to reason with limited information and their saturation points.
3. **Experimental comparison with three other video understanding benchmarks**: We compare Neptune against Video-MME, CinePile and Perception Test using Gemini-1.5-Flash with different numbers of frames, showing the model performance and saturation characteristics on each benchmark.
4. **An additional metric that gives equal weights to different task types**. This allows for a more balanced view of model performance and removes skew of the metric to the most prominent task types.
5. **A breakdown of per-task accuracy**. We have added a table that breaks down performance of all benchmarked models per task and allows detailed insights into each model’s strengths and weaknesses.

Additionally, we will include clarifications that reviewers requested, e.g. the comparison of our pipeline to pipelines used for other benchmarks, and details on our evaluation of the GEM metric. Again, we sincerely thank the reviewers for taking the time to review our paper and providing us with their valuable feedback.

---

### Meta-Review · Area_Chair_JL4t · 2024-12-20

**Metareview:**

This paper introduces NEPTUNE, a benchmark designed to evaluate multimodal understanding of long videos, addressing the limitations of existing datasets that primarily focus on short clips. The paper contributes a semi-automatic pipeline leveraging large video language models (VLMs) and large language models (LLMs) to generate dense, time-aligned video captions and challenging question-answer-decoy (QAD) sets for video segments up to 15 minutes long. Additionally, it proposes a new evaluation metric, GEM (Gemma Equivalence Metric), to assess open-ended responses in a static and reproducible manner. Reviewers commend the dataset's focus on long video reasoning, its innovative annotation pipeline, and the introduction of GEM, which tackles the limitations of prior evaluation metrics. However, concerns about the lack of comparisons with recent benchmarks (e.g., MLVU, Video-MME) and state-of-the-art open-source models (e.g., InternVL, LLaVA-OneVision) are raised. Reviewers also note potential biases in dataset creation due to reliance on VLMs and LLMs, limited novelty in the annotation pipeline, and insufficient evaluation against human judgment for GEM. While the contributions are valuable, the absence of more substantial comparisons and deeper analysis leaves room for improvement.

**Additional Comments On Reviewer Discussion:**

The reviewers appreciate the authors' responses but remain unconvinced on key concerns, maintaining their previous scores. One major issue is the imbalance in question-type distribution within the NEPTUNE benchmark. This could lead to inflated performance for models specializing in specific question types, undermining the benchmark's ability to reflect true model capabilities. While metric adjustments may partially address this, reviewers argue they could compromise robustness and evaluation accuracy for underrepresented tasks. Additionally, concerns persist regarding the semi-automatic benchmark construction process. Reviewers question its scalability advantages, emphasizing that accuracy and diversity should take precedence over scalability for benchmarks. Extending the pipeline to generate high-quality long-video instruction-tuning datasets could better demonstrate its utility and community impact. While some concerns were mitigated, the fundamental issues with question-type balance and the semi-automated approach remain unresolved.

---

### Decision · Program_Chairs · 2025-01-22

Reject